DOI: 10.1038/s41467-018-03237-5　**OPEN**

# Spin-orbit interaction of light induced by transverse spin angular momentum engineering

Zengkai Shao[1], Jiangbo Zhu [2], Yujie Chen [1], Yanfeng Zhang[1] & Siyuan Yu[1,2]

The investigations on optical angular momenta and their interactions have broadened our knowledge of light's behavior at sub-wavelength scales. Recent studies further unveil the extraordinary characteristics of transverse spin angular momentum in confined light fields and orbital angular momentum in optical vortices. Here we demonstrate a direct interaction between these two intrinsic quantities of light. By engineering the transverse spin in the evanescent wave of a whispering-gallery-mode-based optical vortex emitter, a spin-orbit interaction is observed in generated vortex beams. Inversely, this unconventional spin-orbit interplay further gives rise to an enhanced spin-direction locking effect in which waveguide modes are unidirectionally excited, with the directionality jointly controlled by the spin and orbital angular momenta states of light. The identification of this previously unknown pathway between the polarization and spatial degrees of freedom of light enriches the spin-orbit interaction phenomena, and can enable various functionalities in applications such as communications and quantum information processing.

---

[1] School of Electronics and Information Engineering, State Key Laboratory of Optoelectronic Materials and Technologies, Sun Yat-sen University, Guangzhou 510275, China. [2] Photonics Group, School of Computer Science, Electrical and Electronic Engineering and Engineering Maths, University of Bristol, Bristol BS8 1UB, UK. Zengkai Shao and Jiangbo Zhu contributed equally to this work. Correspondence and requests for materials should be addressed to Y.Z. (email: zhangyf33@mail.sysu.edu.cn) or to S.Y. (email: s.yu@bristol.ac.uk)

light waves possess intrinsic spin and orbital angular momentum (SAM and OAM), as determined by the polarization and spatial degrees of freedom of light[1–3]. These two components are separately observable in paraxial beams[4–7], whereas it is well known that fundamentally such a distinction is difficult in light fields with high nonparaxiality and/or inhomogeneity[8–11]. In fact, spin-orbit interactions (SOIs) can be widely observed in light through scattering or focusing[12,13], propagation in anisotropic/inhomogeneous media[14,15], reflection/refraction at optical interfaces[16,17], etc. Notably, the spatial and polarization properties of light are coupled and SOI phenomena must be considered in modern optics dealing with sub-wavelength scale systems, including nano-photonics and plasmonics[18–22]. A variety of novel functionalities utilizing structured light and materials are underpinned by SOI of light, e.g., optical micro-manipulations[23], high-resolution microscopy[24], and beam shaping with planar structures (metasurfaces)[25].

On the other hand, the study of SOI over the past few years has been accompanied by a rising interest in the transverse SAM of light, which has been revealed by recent advances in optics as a new member in the optical angular momentum (AM) family[26–29]. In sharp contrast to the longitudinal SAM predicted by Poynting[1], the transverse SAM exhibits a spin axis orthogonal to the propagation of light[28,30]. Transverse SAM can be typically found in highly inhomogeneous light fields, including surface plasmon polaritons[26], evanescent waves of guided and unguided modes[22,28], and strongly focused beams[31], where longitudinal field components emerge due to the transversality of electromagnetic waves[32]. Light fields possessing transverse SAM can enable various applications in bio-sensing, nano-photonics, etc. More interestingly, transverse spin in evanescent waves also originates from the SOI in laterally confined propagating modes[11], or can be interpreted as the quantum spin Hall effect (QSHE) of light[33–35], and thus gives rise to robust spin-controlled unidirectional coupling at optical interfaces[18,21,22,36–42]. This extraordinary characteristic of transverse SAM results in the breaking of the directional symmetry in mode excitation at any interface

supporting evanescent waves, and can find applications in optical diodes[43], chiral spin networks[44,45], etc.

The ability to simultaneously tailor light fields in the polarization and spatial degrees of freedom via SOI phenomena has allowed for new functionalities in structured light manipulation[46]. Furthermore, combining SOI and transverse SAM control will provide a more versatile platform for processing of light fields in the full AM domain.

Here we present an enrichment of the SOI effects revealed by the engineering of transverse spin in evanescent waves. Our method evolves from a whispering-gallery mode (WGM) resonator-based optical vortex emitter[47,48]. We show that the engineering of transverse spin in the WGM evanescent waves leads to the spin-to-orbital AM conversion in the emitted vortices. This direct interaction between the transverse SAM and intrinsic OAM of light provides a promising pathway toward more sophisticated light manipulation via SOI phenomena. By reversing the emission process, we further demonstrate directional coupling of optical vortices into this integrated photonic circuitry, with the direction of the waveguide modes jointly controlled by the incident spin and orbital AM states, realizing the selective reception of vector vortices without separate polarization and spatial phase manipulation. These results can be used to bring novel functionalities to nano-photonic devices, e.g., encoding and retrieving photonic states in the SAM-OAM space, and provide the guidelines for the design of a nano-photonic chiral interface between traveling and bounded vector vortices.

## Results

**Transverse spin in optical vortex emitter.** The schematic of the platform for the investigation of transverse spin engineering-based SOI is shown in Fig. 1a, where a single-transverse-mode ring resonator is coupled with a two-port access waveguide and embedded with periodic angular scatterers in the inner-sidewall evanescent region of the waveguide. With the sub-wavelength scatterers arranged in a second-order grating fashion, the diffracted first-order light from the evanescent fields of WGMs

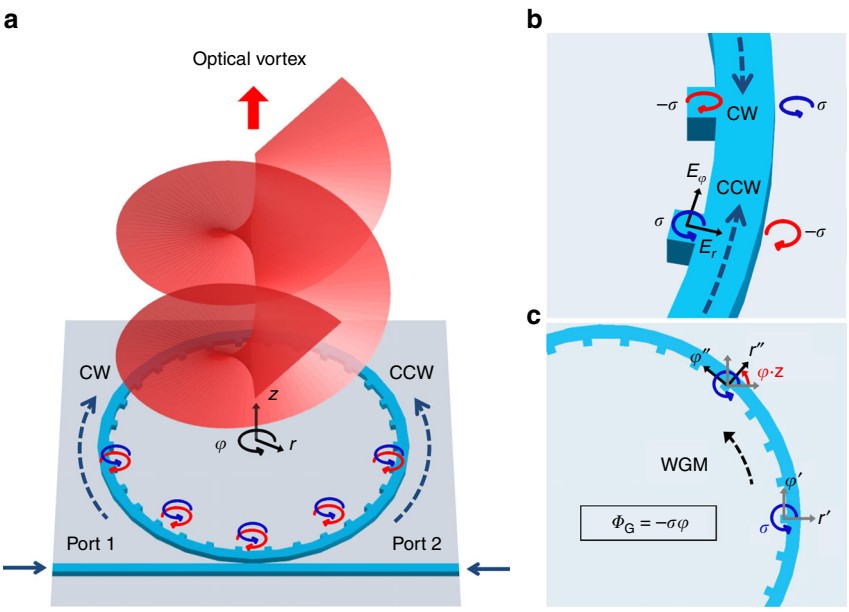

**Fig. 1** Illustration of the concepts. **a** Schematic of the platform for the investigation of transverse spin-induced SOI effect. **b** The clockwise (CW) and counter-clockwise (CCW) WGMs present opposite transverse spins on each side of the resonator. **c** Illustration of the transverse-spin-dependent geometric phase acquired by the vector evanescent wave as the WGM travels around the resonator. For the CCW WGM shown here, a rotation angle of $\varphi \cdot z$ is experienced by the local coordinates from point $(r', \varphi')$ to $(r'', \varphi'')$, and the geometric phase imparted on the evanescent wave with a transverse-spin state $\sigma$ is $\Phi_G = -\sigma\varphi$

collectively produce a vortex beam carrying optical OAM and traveling perpendicular to the resonator plane[47]. In addition, the emitted vortex beams exhibit cylindrically symmetric polarization and intensity distributions, and are thus referred to as cylindrical vector vortices (CVVs)[48,49].

Generally, for the quasi-transverse-electric (quasi-TE) WGMs propagating in the high-index waveguide, a local longitudinal electric component ($E_\varphi$) exists in the sidewall evanescent waves and is in quadrature phase with respect to the radial component ($E_r$; Fig. 1b), as a direct result of the strong lateral confinement and transversality condition[32]. Consequently, the local SAM in the evanescent field exhibits a "transverse" spinning axis in the $z$-direction[50], being orthogonal to the local propagation direction ($+\varphi$ or $-\varphi$) of the WGM. Note that for quasi-TE WGMs, the transverse SAM at the inner and outer sidewalls always has opposite spin directions, and the transverse spin can also be flipped by injecting light from the alternative ports one or two and exciting counter-clockwise (CCW) or clockwise (CW) WGMs, as shown in Fig. 1b.

**Interaction of transverse spin and OAM.** The emission of CVVs from such structures can be generally described in the form of transfer matrices as $\mathbf{E}_{\text{out}} = \mathbf{M}_2 \cdot \mathbf{M}_1 \cdot \mathbf{E}_{\text{in}}$. Assuming the WGM maintains a uniform distribution around the resonator, the generic input light for the matrices is the inner-sidewall evanescent wave and can be written in the locally transverse ($E_r$) and longitudinal ($E_\varphi$) polarization basis. Here the CCW propagating WGM is considered as an example and thus $\mathbf{E}_{\text{in}} \propto e^{ip\varphi}[E_r\ E_\varphi]^{\text{T}}$, where the integer $p > 0$ is the azimuthal mode number and $E_z$ is negligible at the sidewalls[51]. First, the perturbation to WGM evanescent waves induced by the scatterers is expressed by the matrix

$$\mathbf{M}_1 = \begin{bmatrix} W_{rr} & 0 \\ 0 & W_{\varphi\varphi} \end{bmatrix} \cdot e^{i\delta(\varphi)} \quad (1)$$

where $\delta(\varphi) = -q\varphi$ (Supplementary Note 2) is the azimuthal phase acquired by the second-order grating scattering, $q$ is the total number of scatterers around the resonator, and $W_{ii}$ ($i = r, \varphi$) are real numbers quantifying the scatterers' modulation on the strength of the electric components $E_i$ (Supplementary Note 1). Here we define the local transverse-spin state of the perturbed evanescent wave $|\mathbf{M}_1| \cdot \mathbf{E}_{\text{in}} \propto e^{ip\varphi}[E_{rr}\ E_{\varphi\varphi}]^{\text{T}}$, where $E_{ii} = W_{ii}E_i$, as (see Supplementary Note 3 for details)

$$\sigma = \frac{i\left(E_{rr}E_{\varphi\varphi}^* - E_{rr}^*E_{\varphi\varphi}\right)}{|E_{rr}|^2 + |E_{\varphi\varphi}|^2} \quad (2)$$

Note that $\sigma$ ($|\sigma| \leq 1$) is a real number as $E_{\varphi\varphi}$ and $E_{rr}$ always oscillate in quadrature with each other at the sidewalls[51], and it directly characterizes the spatial density of transverse spin[6,11]. For the transverse SAM of left (right)-handed spin here, $\sigma > 0$ ($< 0$). In addition, the vector fields of CCW WGMs traveling along the resonator experience a rotation of local coordinates ($r, \varphi$) with respect to the global reference frame ($x, y$) as $\varphi \cdot \mathbf{z}$ (see Fig. 1c, and $\mathbf{z}$ is unit vector), which can also be described by the matrix

$$\mathbf{M}_2 = \begin{bmatrix} \cos\varphi & -\sin\varphi \\ \sin\varphi & \cos\varphi \end{bmatrix} \quad (3)$$

By applying the transfer matrices $\mathbf{M}_1$ and $\mathbf{M}_2$, the Jones vector of the output CVV becomes

$$\mathbf{E}_{\text{out}} \propto \frac{1}{2}\left\{ \sqrt{1+\sigma}\,e^{i(l_{\text{TC}}-1)\varphi}\begin{bmatrix} 1 \\ i \end{bmatrix} + \sqrt{1-\sigma}\,e^{i(l_{\text{TC}}+1)\varphi}\begin{bmatrix} 1 \\ -i \end{bmatrix} \right\} \quad (4)$$

where $l_{\text{TC}} = p - q$ is defined as the topological charge (TC)[47]. Here the Jones vector is formulated in the global reference frame with the $x$- and $y$-polarization basis (i.e., $[E_x\ E_y]^{\text{T}}$). The above interaction between the vector evanescent wave and grating can further be elucidated with the spatial phase acquired by the CVV. Pancharatnam phase[52,53], extensively used for comparing the phase between light fields of different polarization states, is here used to describe the spatial (angular) phase variation in CVVs shown in Eq. (4) (Supplementary Note 6)

$$\Phi_{\text{P}} = l_{\text{TC}}\varphi - \sigma\varphi \quad (5)$$

where $l_{\text{TC}}\varphi = p\varphi + \delta(\varphi)$ is the scattering phase solely resulted from the first-order diffraction of the grating. Meanwhile, the second term, $\Phi_{\text{G}} = -\sigma\varphi$, has a pure geometric nature and arises from the rotation of local transverse-spin state while CCW WGMs travel around the resonator, as described by Eq. (3). Here $\Phi_{\text{G}}$ differs from all the previously discussed geometric phases of light that can be identified either in artificial anisotropic structures[52] or light beams of curvilinear trajectories[15], and originates essentially from the coupling between the transverse SAM of guided light and the rotation of light's path.

On the other hand, it is straightforward to obtain the $z$-components (along the CVV propagating direction) of SAM and OAM carried by each photon in CVVs as (Supplementary Note 5)

$$S_z = \sigma\hbar, \quad L_z = (l_{\text{TC}} - \sigma)\hbar \quad (6)$$

where $\hbar$ is the reduced Planck constant. Notably, one can find that the transverse-spin-dependent geometric phase $\Phi_{\text{G}} = -\sigma\varphi$ still complies with the unified form of geometric phase of light $\Phi_{\text{G}} = -\int \mathbf{S} \cdot \mathbf{\Omega}_\varphi\, d\varphi$[11], and here $\mathbf{S} = S_z\hbar^{-1} \cdot \mathbf{z}$ is the SAM vector and $\mathbf{\Omega}_\varphi = +\mathbf{z}$ is the angular velocity of coordinate rotation for CCW WGMs (Fig. 1c). For CW WGMs, $\mathbf{\Omega}_\varphi = -\mathbf{z}$ and $\Phi_{\text{G}} = \sigma\varphi$.

Meanwhile, the $z$-component of total AM (TAM) in the emitted CVVs ($J_z = L_z + S_z = l_{\text{TC}}\hbar$) is found to be conserved with the given WGM azimuthal mode order $p$ and grating number $q$, regardless of the transverse-spin state. This is attributed to the rotationally symmetric "anisotropy" orientation of the scatterer group[54], and consequently the net transfer of AM between the WGMs (carrying TAM of $p\hbar$ per photon) and device is constantly $-q\hbar$. More significantly, a direct conversion between the intrinsic OAM in optical vortices and transverse SAM in evanescent waves can be identified in Eq. (6). By engineering the transverse-spin state $\sigma$ and consequently the transverse-spin-dependent geometric phase $\Phi_{\text{G}}$, the OAM state of a CVV can be modulated and partially converted with SAM. This effect can also be reasonably expected in other systems, where the geometric phase stemming from the transverse spin rotation contributes to the spatial phase reshaping of highly confined light.

In addition, the left- and right-hand circular polarization (CP) vortices in Eq. (4) possess the TCs of $l_{\text{TC}} - 1$ and $l_{\text{TC}} + 1$, respectively. The composition of this "superposition" is subject to the transverse-spin state of WGM evanescent wave. Particularly, when the polarization at the grating scatterer locations reaches one of the CP states (i.e., $\sigma = \pm 1$), this superposition reduces to a single CP scalar vortex state with a single OAM eigenstate ($l = l_{\text{TC}} \mp 1$).

It should be mentioned that by exciting WGMs from the alternative waveguide ports or scattering the evanescent waves on

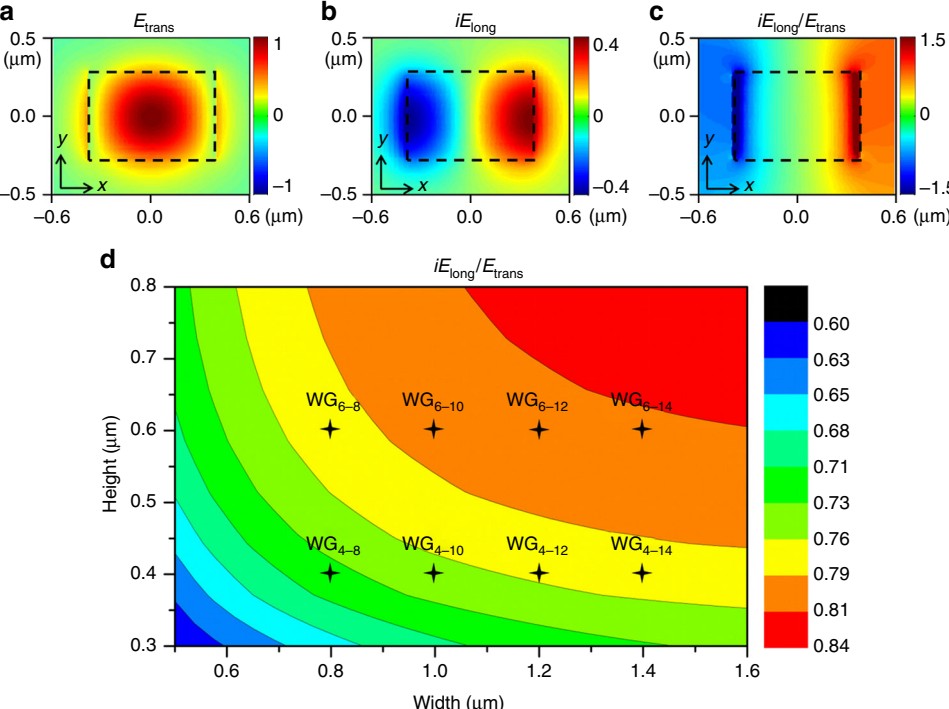

**Fig. 2** Numerically calculated field component distributions and their dependence on waveguide dimensions. **a** The cross-sectional field distribution of the transverse component $E_{\text{trans}}$ of the fundamental quasi-TE mode in a $SiN_x$ waveguide, and the dashed rectangular indicates the waveguide of 0.8 µm width and 0.6 µm height. The results in **b** and **c** are obtained with the same waveguide mode. **b** The field distribution of the longitudinal component (multiplied with the imaginary unit) $iE_{\text{long}}$. **c** The distribution of the component ratio $iE_{\text{long}}/E_{\text{trans}}$ over the waveguide cross section and evanescent region. **d** The contour map of the ratio $iE_{\text{long}}/E_{\text{trans}}$ over variable waveguide dimensions. Among all the waveguide designs calculated, eight waveguide dimensions marked in the map are employed for device fabrication and characterization, consisting of two different heights (0.4 and 0.6 µm) and four widths (0.8, 1.0, 1.2, and 1.4 µm) as indicated in the subscripts

**Table 1 Design parameters of the fabricated devices**

| Sample | WG$_{4-8}$ | WG$_{4-10}$ | WG$_{4-12}$ | WG$_{4-14}$ | WG$_{6-8}$ | WG$_{6-10}$ | WG$_{6-12}$ | WG$_{6-14}$ |
|---|---|---|---|---|---|---|---|---|
| Waveguide height (µm)[a] | 0.4 | 0.4 | 0.4 | 0.4 | 0.6 | 0.6 | 0.6 | 0.6 |
| Waveguide width (µm)[a] | 0.8 | 1.0 | 1.2 | 1.4 | 0.8 | 1.0 | 1.2 | 1.4 |

[a]These parameters apply both to the ring waveguide and access waveguide

the other side of resonator waveguide, the sign of the transverse spin will be flipped, and using the alternative waveguide port to excite CW propagating WGMs will also reverse the sign of $l_{\text{TC}}$ (Supplementary Note 2). Nevertheless, the general SOI phenomena and mode decomposition described in Eqs. (4) and (6) still hold.

**Transverse spin engineering**. The transverse-spin state $\sigma$ in the waveguide evanescent wave is generally subject to the ratio of longitudinal ($E_{\text{long}}$, along waveguide surface) and transverse ($E_{\text{trans}}$, normal to waveguide surface) field components, $iE_{\text{long}}/E_{\text{trans}}$, in the evanescent region (Supplementary Note 3). In contrast to the evanescent waves of WGMs in bottle micro-resonators[55] and unbounded evanescent waves at optical interfaces[26], where this ratio is largely determined by the refractive index (RI) contrast and the incident angle of light, the transverse spin of evanescent waves in highly confined waveguide modes is also significantly altered by the lateral confinement conditions, especially the waveguide core dimensions. By modifying the mode profile of the transverse component in the core and its spatial derivative at the waveguide boundaries, the magnitude of the

ratio $iE_{\text{long}}/E_{\text{trans}}$ can be engineered[56]. In other words, by tailoring the waveguide geometry and consequently the vector components of modes, $\sigma$ can be adjusted and thus enable the engineering of transverse spin in evanescent waves[57].

As an example, the cross-sectional maps of the fundamental quasi-TE mode components in a straight silicon nitride ($SiN_x$) waveguide (surrounded by air and placed on a $SiO_2$ substrate) is depicted in Fig. 2, where the dashed rectangles indicate the waveguide core of 0.6 µm in height and 0.8 µm in width. Apart from $E_{\text{trans}}$ (Fig. 2a), a strong $E_{\text{long}}$ at the core-cladding interface can also be observed in $\pm\pi/2$ phase difference to $E_{\text{trans}}$, as shown in Fig. 2b. The map of the ratio $iE_{\text{long}}/E_{\text{trans}}$ is also plotted in Fig. 2c, and outside the waveguide sidewalls it remains almost constant in the decaying evanescent wave, as both components decay at the same rate. More importantly, a contour map of this ratio is plotted in Fig. 2d, in which a variable ratio of the two components can be observed over various waveguide dimensions. Variable transverse-spin state in waveguide evanescent wave can thus be achieved with routine waveguide design[56]. The eight waveguide dimensions we choose for experimental investigation are marked in the map, and their parameters are listed in Table 1,

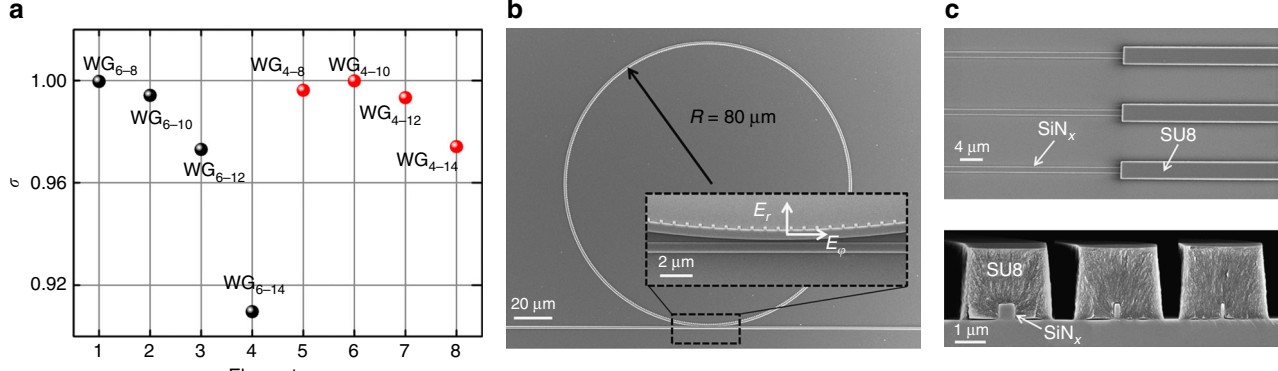

**Fig. 3** Calculated transverse-spin states of all designed devices and SEM images of fabricated device WG$_{6-8}$. **a** Calculated transverse-spin states in the evanescent region of all eight sample devices. **b** SEM image of the device WG$_{6-8}$. The inset shows a close-up of the coupling section between the access waveguide and the resonator. **c** Top: junction point of the tapered coupler consisting of a tapered SiN$_x$ waveguide and a SU8 waveguide. Bottom: cross-section views at various positions of the tapered coupler. The minimum width of the SiN$_x$ taper (shown in the right-hand side image) is 130 nm

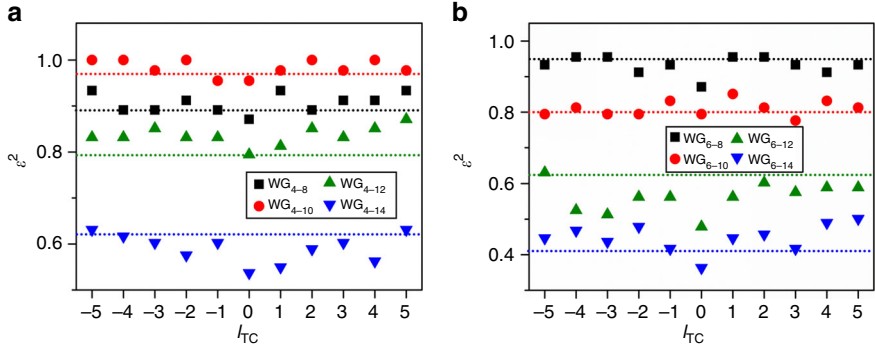

**Fig. 4** Characterization of average polarization state in CVVs. **a**, **b** Measured squared polarization ellipticity $\varepsilon^2$ (solid markers) of the CVVs from the devices of height 0.4 and 0.6 μm, respectively. The prediction of $\kappa^2$ from numerical calculations is plotted with dashed lines, and the measured and calculated results for the same device are marked in the same color

where the subscripts $w$ and $h$ in the notation for waveguide sample WG$_{w-h}$ indicate the width and height of waveguide (in the unit of 100 nm), respectively. SiN$_x$ waveguide is employed for its moderate RI (RI ~ 2.01) so that typically a larger range of transverse-spin state can be accessed than other materials (e.g., silicon with RI ~ 3.5).

Intuitively, the simplest geometry of the scatterers for projecting the evanescent-field transverse-spin information into far-field with high fidelity is a spherical Rayleigh particle[22]. However, the fabrication of sub-micron spherical scatterers, whether of metallic or dielectric nature, placed uniformly in the evanescent region along the resonator is rather challenging. Here we utilize square cuboid structures (as shown in Fig. 1b), attached to the inner sidewall and of the same height as waveguide, as Rayleigh scatterers, allowing for both one-step etching in fabrication and highly uniform scattering of evanescent waves along the entire resonator (see Supplementary Note 1).

For the eight sample devices, the ring radius $R = 80$ μm. For each device, the number of scatterers $q = 517$, and each scatterer has a planar size 100 nm by 100 nm. The gap between the access waveguide and ring resonator is fixed at 200 nm. The calculated transverse-spin state of all sample devices over the scatterer region is shown in Fig. 3a (see Methods). $\sigma > 0$ holds for all cases with WGMs excited by injecting light into Port 1 and the evanescent wave at the inner sidewall is left-hand elliptical-polarized. In particular, near-circular transverse spin is expected from the devices WG$_{6-8}$ and WG$_{4-10}$ with $\sigma \approx 1$. Some scanning electron microscope images of device WG$_{6-8}$ are shown in Fig. 3b, c.

**Polarization and transverse-spin state characterization.** First, the average "cylindrical" polarization ellipticity of the CVVs is measured to show the overall effect of near-field transverse spin on the polarization of far-field CVVs. The polarization of CVVs varies in space but exhibits a cylindrical symmetry with respect to the propagation axis[48], and therefore here the components $E_r$ and $E_\varphi$ are measured to characterize the average global ellipticity in the cylindrical basis (i.e., $\varepsilon = |E_r|/|E_\varphi|$ or $|E_\varphi|/|E_r|$), and compared with the calculated near-field component ratio $\kappa = iE_{\varphi\varphi}/E_{rr}$ defined in the same basis (see Methods). A radial polarization convertor is used to convert $E_r$ and $E_\varphi$ in far-field CVVs into $x$- and $y$-polarized fields, respectively[58], and the power of these two components ($P_r$ and $P_\varphi$) is then recorded for $\varepsilon^2$ calculation ($\varepsilon^2 = P_r/P_\varphi$ or $P_\varphi/P_r$) (see Supplementary Note 7).

The measured $\varepsilon^2$ in CVVs of various $l_{TC}$ from all devices is shown in Fig. 4 as solid markers, while the corresponding predicted $\kappa^2$ of each device is plotted as the dashed line in the same color. Overall, the measured $\varepsilon^2$ exhibits high uniformity over all $l_{TC}$. CVVs of a wide range of spin states ($\varepsilon^2$ from ~0.4 to ~1.0) is obtained with various waveguide designs, and the agreement between the $\varepsilon^2$ and $\kappa^2$ shows a definitive correspondence from the transverse-spin state in guided evanescent waves to the polarization in emitted vortices. Particularly, near-CP ($\varepsilon \approx 1$) CVVs are observed with devices WG$_{4-10}$ and WG$_{6-8}$, indicating that the reduced superposition of single spin-orbital eigenstate vortices predicted by Eq. (4) can be reached.

The results above also indicate that the transverse-spin state once bounded in the near-field now manifests itself as the

longitudinal (i.e., along $z$-direction) spin state in the propagating CVVs. Therefore, by performing the spatially resolved Stokes polarimetry to the near-field image of CVVs, the local transverse-spin state distribution in near-field CVVs can be revealed (see Supplementary Note 7). With the Jones vector shown in Eq. (4), the normalized Stokes parameters as a function of the azimuthal coordinate can be obtained as[59]

$$S_1 = \sqrt{1-\sigma^2} \cos 2\varphi, S_2 = \sqrt{1-\sigma^2} \sin 2\varphi, S_3 = \sigma \quad (7)$$

For a device of a larger $|\sigma|$, the trajectory of the Stokes vector [$S_1$, $S_2$, $S_3$] on the Poincare sphere circles the pole twice at a higher latitude parallel to the equator. For $|\sigma| = 1$, the circle contracts to a single point at the poles, producing a CP CVV.

The measured Stokes parameters of near-field CVVs are depicted in Fig. 5, in which the results of $l_{TC} = +4$ CVVs from the devices $WG_{6-8}$, $WG_{6-10}$, $WG_{6-12}$, and $WG_{6-14}$ are shown. The curves in each case of Fig. 5a–d show the dependence of Stokes parameters on the $\varphi$-coordinate, including the predictions from Eq. (7) (solid curves) and the measured values (dots). First, the dots of each measured $S_1$ or $S_2$ complete the oscillation of two full sinusoidal cycles around the resonator, following the theory predictions (curves). And thus, the rotation of near-field transverse-spin state shown in Fig. 1c is visualized, thereby indicating the existence of the geometric phase predicted in Eq. (5). On the other hand, the modulation effect of waveguide geometry on the local transverse-spin state in evanescent waves can be validated, by comparing the amplitude of $S_1$ ($S_2$) oscillation from the four devices in Fig. 5. Sample $WG_{6-8}$, with which $S_1$ and $S_2$ hardly oscillate, produces the largest local polarization ellipticity of the four in the evanescent region. More discussion on the results of Stokes polarimetry is provided in Supplementary Note 9.

**Transverse-spin-induced SOI.** The OAM component carried by CVVs is measured to verify the transverse-spin-induced spin-to-orbital conversion predicted by Eq. (6) (see Supplementary Notes 7 and 8 for characterization method of OAM state and emission spectrum from devices). The measured OAM spectra for the CVVs from the devices $WG_{6-8}$, $WG_{6-10}$, $WG_{6-12}$, and $WG_{6-14}$ are plotted in Fig. 6. In close agreement with the theory, each OAM spectrum (row) of CVV with $l_{TC}$ contains two dominant peaks at $l_{TC} - 1$ and $l_{TC} + 1$, carried by the constituent left- and right-hand CP vortices, respectively. The intensities of all spurious modes are <0.03. Each CP vortex can thus be confirmed as possessing a TAM of $l_{TC}\hbar$, and this experimentally validates the overall TAM in each CVV is preserved as $l_{TC}\hbar$ regardless of waveguide geometries. More importantly, the average SAM in each CVV is subject to the near-field transverse spin ($S_z = \sigma\hbar$), as shown in Fig. 5. Therefore, the remarkable transverse-spin-dependent SOI effect is revealed, as the OAM component carried by CCVs can be partially derived out of the transverse SAM in the evanescent waves.

A direct and useful manifestation of this effect is that the relative intensities of the two dominant peaks, i.e., the two constituent CP vortices, can be changed by modifying $\sigma$. For example, the normalized intensities of the left- and right-hand CP vortices from $WG_{6-8}$ are around 0.93 and 0.07, respectively, while for $WG_{6-14}$ they account for about 0.62 and 0.36 of the total intensity, respectively. This variable superposition of AM states in CVVs provides a viable pathway for information encoding in the spin-orbit space. Another implication of this SOI effect is that a vortex should appear even when $l_{TC} = 0$ but $\sigma \neq 0$ (exemplified by the square in the yellow box in Fig. 6a); that is, without introducing any spatial phase gradient that has been inherent to many optical vortex generation techniques[5]. This purely

transverse-spin-derived vortex essentially originates from the spatially varying "anisotropy" of the gratings and the rotational symmetry of vector WGMs. In other words, this is an interesting demonstration of optical vortex generation controlled by the QSHE of light[33], and the spin state in the edge modes stemming from the intrinsic SOI at optical interfaces can thus be manipulated for spatial light modulation via the "extrinsic" SOI in anisotropic structures[11].

**Spin-orbit-controlled unidirectional coupling.** With the principle of reciprocity, this device can also be used for detection of AM states in an incident CVV beam[60]. The ring resonator supports the degenerate CW and CCW WGMs at each resonance wavelength $\lambda_{res}$, giving rise to the emission of two CVVs of opposite TCs $l_{res} = \pm(p-q)$ (Supplementary Note 2). Meanwhile, these two WGMs exhibit opposite $\sigma_{res}$ in the inner-side evanescent fields, and therefore the two CVVs show exactly opposite spin and orbital AM states, i.e., $\langle \pm\sigma_{res}, \pm l_{res} \mp \sigma_{res} \rangle$. Here the notation $\langle s_z, l_z \rangle$ is used to indicate the SAM ($s_z$) and OAM ($l_z$) states of light. Presumably, when receiving at $\lambda_{res}$, this device should couple these two CVVs into the resonator as two counter-propagating (i.e., CW and CCW) WGMs and guide their power to the respective access ports. Generally, the incident light for ideal reception should carry the identical spin and orbital AM states as the emitted CVVs, while exhibiting cylindrical symmetry in intensity and polarization profiles. For simplicity, the following analysis and demonstration are exemplified by CVVs of reduced spin states, i.e., $\sigma_{res} = \pm 1$. The schematic of this directional coupling of CVV modes is shown in Fig. 7a. First, the local coupling of waveguide modes via evanescent wave at the scatterers is governed by the spin-direction locking effect[21,22]. Namely, the sign of local input spin state $\sigma_{in}$ ($=\pm 1$) dictates the circulating direction of coupled light, i.e., CW or CCW. Additionally, the excited modes must follow the phase-matching condition for the resonance as WGMs (Supplementary Note 2), which can be written here as $l_{in} + \sigma_{in} = l_{res}$ and $l_{in}$ is the OAM state of incident vortex. Remarkably, the effect of unidirectional coupling into guided modes jointly controlled by spin and orbital AM states of light is thus suggested. This advanced spin-direction locking effect incorporates the spatial degree of freedom, using the close-loop waveguide for filtering in the OAM space. Such spin-orbit jointly controlled coupling provides a potential solution for spin and orbital AM state detection, avoiding the separate manipulations on these two degrees of freedom.

To verify this prediction, experimental investigation using the device $WG_{4-10}$ of near-CP transverse-spin state ($\sigma_{res} \approx \pm 1$) is performed, and the results of incident CVVs with 3 spin states ($\sigma_{in} = 0, \pm 1$) are shown in Fig. 7b–d. With each $\sigma_{in}$, CVVs at five resonance wavelengths of $WG_{4-10}$ are prepared and illuminated on the device (see Supplementary Note 10 for details). At each wavelength, the OAM value $l_{in}$ of incident CVV is changed across 11 states from $-5$ to $+5$. The wavelengths are selected from the emission spectrum of the device (cf. Supplementary Note 8), such that the at these five $\lambda_{res}$ the CVVs emitted from this device exhibit $l_{res} = \pm 4, \pm 2, 0, \mp 2, \mp 4$, respectively (note that at each $\lambda_{res}$ light injected from Port 1 and Port 2 leads to $l_{res}$ of opposite signs).

In Fig. 7b–d, each blue (red) bar indicates the normalized power $P_1$ ($P_2$) received out of Port 1 (2) with the given illuminance of states $\langle \sigma_{in}, l_{in} \rangle$. The first distinctive observation is that $P_1$ ($P_2$) is universally negligible with incidence of $\sigma_{in} = +1$ ($-1$), indicating the breaking of the mirror symmetry (with respect to the radial axis at each grating point) in mode coupling. This is in accordance with the predefined $\sigma_{res} \approx -1$ ($+1$) when inputting via Port 1 (2) and the underlying prediction from spin-

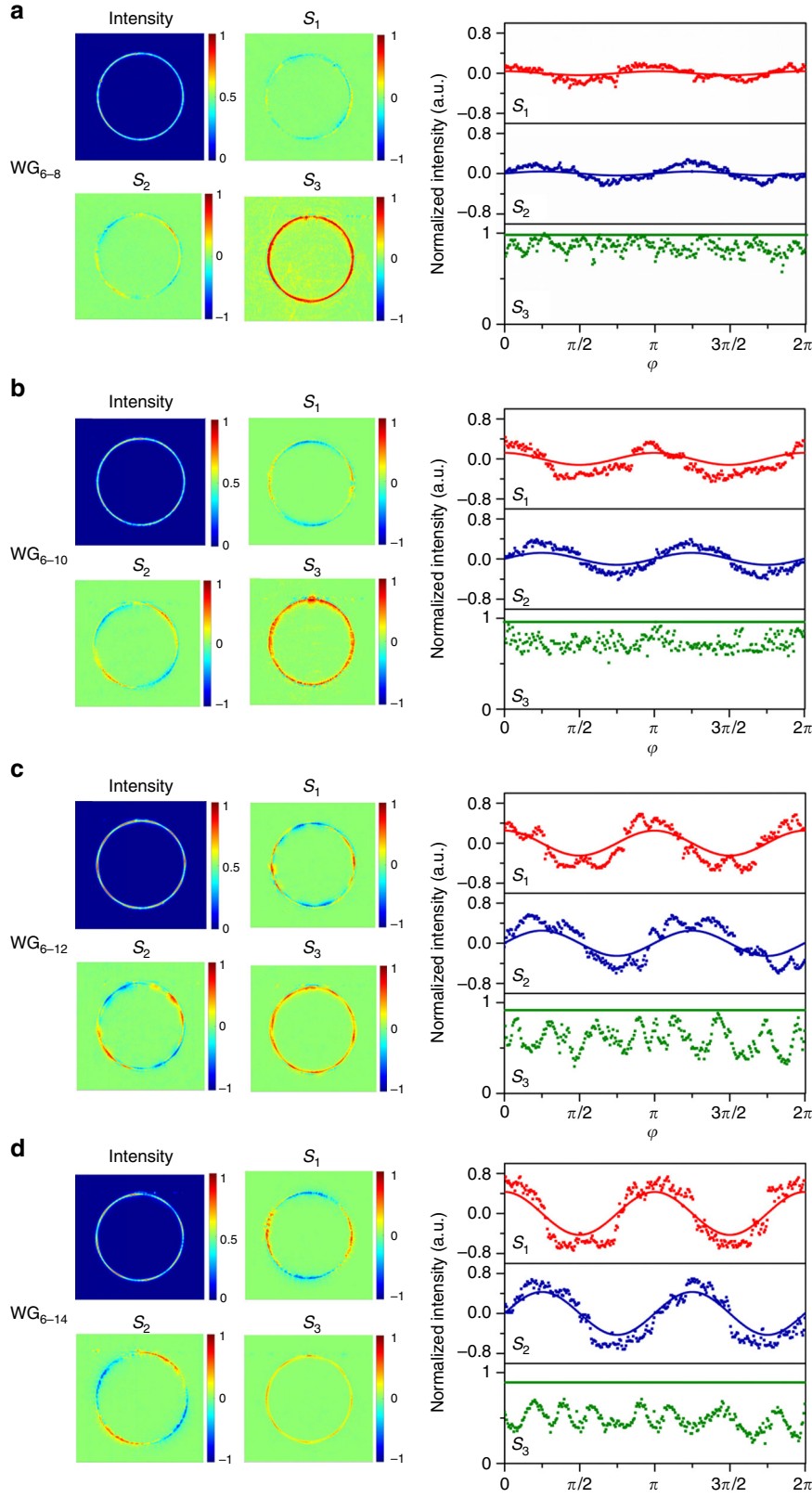

**Fig. 5** Stokes polarimetry of near-field polarization of CVVs. **a–d** Measured two-dimensional maps of near-field Stokes parameters and the comparison with theoretical prediction for devices WG$_{6-8}$, WG$_{6-10}$, WG$_{6-12}$, and WG$_{6-14}$, respectively. In each case, the four maps show the measured near-field intensity profile of the device with $I_{TC} = +4$, and the normalized Stokes parameters $S_1$, $S_2$, and $S_3$, respectively. The curves in each case show the comparison between the measured results (dots) sampled from the maps and the corresponding prediction (solid curves) from Eq. (7). For each set of measured data, 288 pixels intersecting with the circle of 80 μm radius along the azimuthal direction (φ) from 0 to 2π are sampled from the corresponding map. For each solid curve, the data are calculated by substituting the transverse-spin state σ from Fig. 3a into Eq. (7)

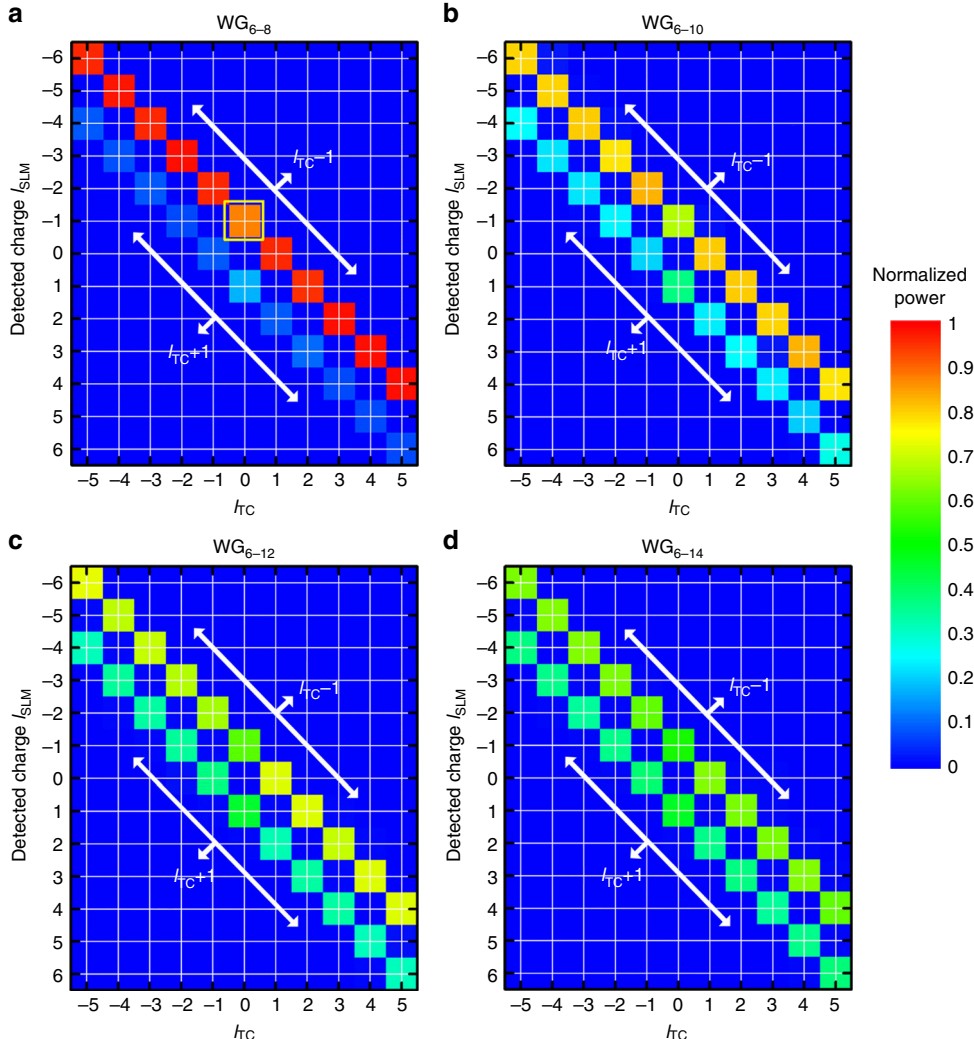

**Fig. 6** Characterization of OAM components in CVVs. **a–d** The measured OAM spectra for the devices $WG_{6-8}$, $WG_{6-10}$, $WG_{6-12}$, and $WG_{6-14}$, respectively. For each device, the wavelengths of $l_{TC} = -5$ to $+5$ are considered, and each column represents a spectrum of measured OAM components with the corresponding $l_{TC}$

direction locking[21,22]. With the incidence of a polarization state other than $\sigma_{in} = \pm 1$, however, light is coupled to the both ports and the resultant ratio of $P_1$ and $P_2$ is determined by the relative intensity of left- and right-hand CPs in the incident CVV. For example, with the incident linear polarization ($\sigma_{in} = 0$) as an equal superposition of two CPs, $P_1$ and $P_2$ show comparable values (Fig. 7c). Moreover, the coupling strength is further subject to the incident OAM state $l_{in}$. For example, when measuring at Port 2 with $\lambda_{res} = 1578.61$ nm ($l_{res} = +4$) and $\sigma_{in} = +1$, a single dominant $P_2$ peak appears only at $l_{in} = +3$ among all 11 states (Fig. 7b), as predicted by the phase-matching condition $l_{in} = l_{res} - \sigma_{in}$, while its counterpart $P_1$ peak at the same $\lambda_{res}$ ($l_{res} = -4$) can only be observed with incidence of $\langle -1, -3 \rangle$ (Fig. 7d). As a result of this spin-orbit-controlled coupling, the recorded power with incidence of $\sigma_{in} = \sigma_{res}$ explicitly presents the identity-matrix-like distribution (Fig. 7b, d), while the arbitrary incidence of $\sigma_{in} \neq \sigma_{res}$ generally leads to a "cross" matrix superimposed by the two identity matrices.

This highly directional and selective coupling, determined by the spin and orbital AM state $\langle \sigma_{in}, l_{in} \rangle$, is a higher-order phenomenon with respect to the basic spin-controlled coupling via evanescent waves, as both the spatial and polarization

properties of light must be taken into account. This effect allows for a robust manipulation of light on the micron-scale using both the spin and orbital degrees of freedom, e.g., encoding and retrieving information, without the necessity of separate controls on polarization and spatial phase profile.

## Discussion

To sum up, we have identified and demonstrated the direct interplay between the intrinsic OAM and the transverse spin of light. This SOI effect originates from the manipulation of local transverse-spin-dependent geometric phase by artificially introducing a closed-loop waveguide and sub-wavelength scatterers with rotational symmetry. Engineering the local transverse spin by tailoring waveguide dimensions then controls the global spin-to-orbital conversion in the generated optical vortices.

Our results have both fundamental and applied importance. The interaction between the intrinsic OAM and transverse spin of light is an integral but thus far missing part of the rich SOI phenomena. The phenomenon discovered here builds one more pathway between the polarization and spatial degrees of freedom of light, which could provide nano-photonic technologies with additional tools of light manipulation at the sub-wavelength scale

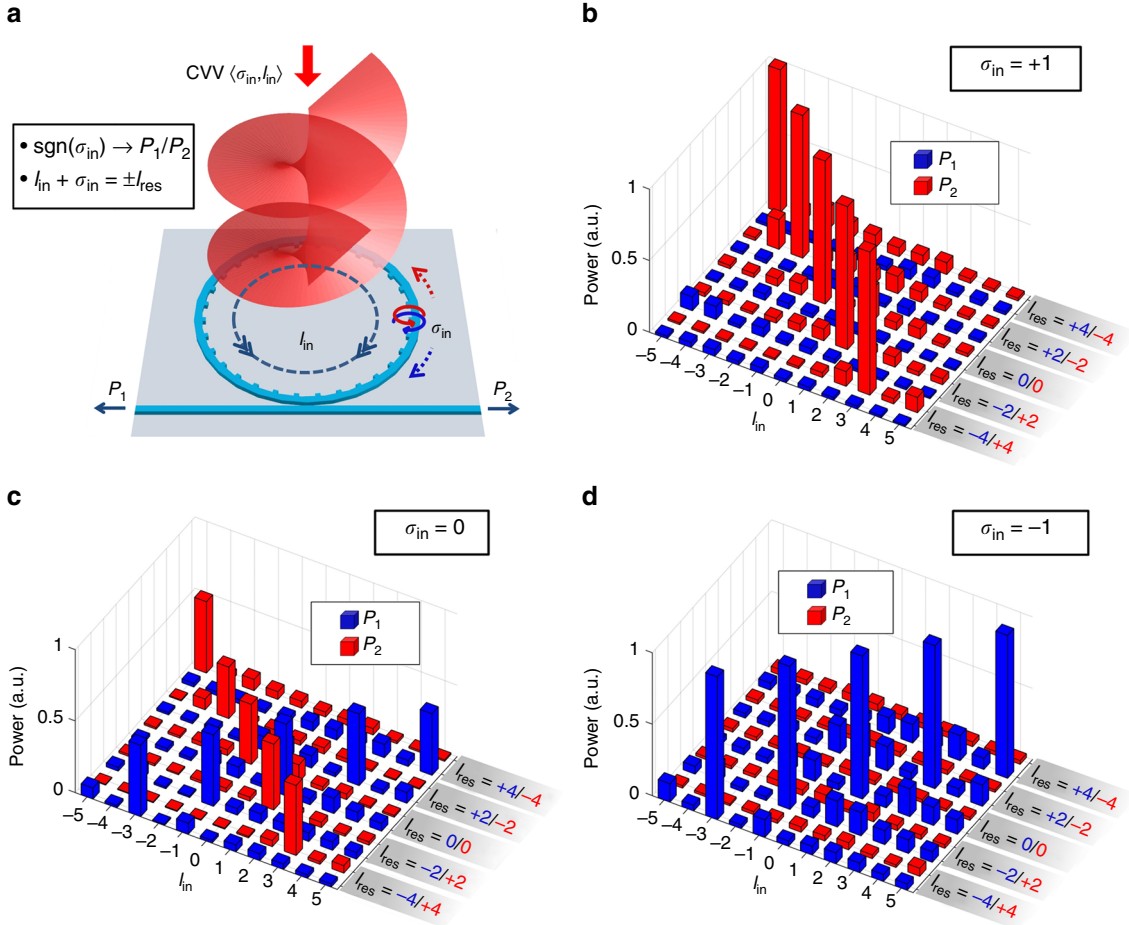

**Fig. 7** Proof-of-principle demonstration of spin-orbit jointly controlled unidirectional coupling of waveguide modes. **a** Schematic of platform for receiving CVVs via spin-orbit unidirectional coupling effect. For devices of unity transverse-spin states in the evanescent region ($\sigma_{res} = \pm 1$), the propagation direction of coupled light in the waveguide, or the ratio of received power at the two ports $P_1/P_2$, is first subject to the sign of incident SAM state, sgn($\sigma_{in}$). Meanwhile, at a given wavelength ($\lambda_{res}$), the incident spin-orbit states $\langle \sigma_{in}, l_{in} \rangle$ must obey the phase-matching condition $l_{in} + \sigma_{in} = l_{res}$ for high-efficiency coupling. **b–d** Received power at Port 1 ($P_1$, blue bars) and Port 2 ($P_2$, red bars) of device $WG_{4-10}$ when the SAM state of incident wave $\sigma_{in} = +1$, 0, and $-1$, respectively. For each $\sigma_{in}$, CVVs at 5 resonance wavelengths of $WG_{4-10}$ (1578.61, 1583.11, 1587.59, 1592.11, and 1596.66 nm) and 11 OAM states ($l_{in} = -5$, $-4$, ..., and $+5$) are illuminated on the device. At these five wavelengths, the CVVs emitted from the device would carry the topological charges of $l_{res} = \pm 4$, $\pm 2$, 0, $\mp 2$, $\mp 4$, respectively, when injecting light into Port 1 (2), as marked in blue (red) on the $l_{res}$ axis in **b–d**. All measured power has been calibrated with respect to the lensed fiber coupling loss (Supplementary Note 10), and all data are normalized to the highest value in each $l_{res}$ group

and of information transfer over more degrees of freedom. The resulting effects, e.g., the variable superposition of spin-orbit states in optical vortices, may find applications in optical quantum information processing. The spin-orbit jointly controlled directional coupling can be used to operate on the eigenstates involving both AM components, so that the device considered here can be regarded as a prototype of a planar spin-orbit-controlled gate that interfaces propagating and bounded photons of two-dimensional entanglement. Better performance (e.g., power efficiency and AM state purity) can be brought about by further device design and optimization. The demonstrated interaction should also exist in other systems that support evanescent modes, including surface plasmon polaritons, which can significantly miniaturize the elements.

## Methods

**Numerical simulation of transverse-spin state**. The transverse-spin state in the evanescent wave of micro-ring WGMs is numerically evaluated with a FDE solver (Lumerical Solutions, Inc.). First, the resonance of the two cylindrical field components ($E_r$ and $E_\varphi$) around one grating region is calculated, assuming a $SiN_x$ ring resonator of $R = 80$ μm (air cladding and $SiO_2$ substrate) and the WGM excited by

injection from Port 1. Then, the power ($P_{rr}$ and $P_{\varphi\varphi}$) of scattered fields ($E_{rr}$ and $E_{\varphi\varphi}$) is calculated by numerically integrating the intensities of $E_r$ and $E_\varphi$, respectively, over the grating region, and thus the ratio of the two components $\kappa = iE_{\varphi\varphi}/E_{rr} = (P_{\varphi\varphi}/P_{rr})^{0.5}$. Subsequently, the transverse-spin state is evaluated using its dependence on this ratio (see Supplementary Note 3)

$$\sigma = -\frac{2\kappa}{1 + \kappa^2}$$

**Fabrication**. The $SiN_x$ waveguide layers are first deposited on a 5-μm oxidized <100> silicon wafer using inductively coupled plasma chemical vapor deposition system (Plasmalab System 100 ICP180, Oxford). The device structures are defined in a 450-nm-thick negative resist using electron-beam lithography (EBPG5000 ES, Vistec). Reactive-ion-etch (RIE, Plasmalab System 100 RIE180, Oxford Instruments) with a mixture of $CHF_3$ and $O_2$ gases is applied to etch through the waveguide layer to form the device. An inverse taper combined with a SU8 waveguide is used as the coupler between external optical fiber and the access waveguide.

**Experimental setups**. Experimental setups for device characterizations (including polarization ellipticity and Stokes polarimetry), SOI measurement, and spin-orbit-controlled unidirectional coupling are shown and explained in Supplementary Notes 7 and 10.

**Data availability**. The data that support the findings of this study are available from the corresponding authors upon request.

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

## Acknowledgements

SYSU is supported by National Basic Research Program of China (973 Program) (2014CB340000), National Natural Science Foundations of China (61490715, U1701661, 11774437, 61323001, and 11690031), National Key Research and Development Program of China (2016YFB0402503), and Science and Technology Program of Guangzhou (201707020017). UoB is supported by European Union Horizon2020 project ROAM.

## Author contributions

J.Z., Y.Z., and S.Y. conceived the idea. Z.S. performed the simulations, and designed and fabricated the devices. J.Z. performed the theory analysis and designed the experiment. Z.S. and J.Z. performed the experiments, analyzed the data, and interpreted the results.

Y.Z., Y.C., and S.Y. supervised the project. J.Z. wrote the manuscript. All authors contributed to the discussions and writing of the manuscript.

## Additional information

**Competing interests:** The authors declare no competing financial interests.

