## [Peer Review File · Nature Communications]

Reviewers' comments:

Reviewer #1 (Remarks to the Author):

Spin-orbit interaction of light induced by transverse spin angular momentum engineering
Zengkai Shao et al.

The authors study an integrated optical structure capable of generating vortex beams from a guided optical waveguide, via a circular resonator with indentations, very similar to Ref. [41] in structure and in physics. However, the authors provide a crucial insight and convincingly demonstrate a novel phenomena. While the authors of Ref. [41] consider their vortex beams as being azimuthally polarized, neglecting other components of the electric field in the guided modes, in the present work the authors take into full account the different components of the electric field in the waveguide, and crucially, their phase relation, to show that the vortex beam will have different circular polarization components (spin angular momentum) on top of their vortex character (orbital angular momentum). The ratio of their amplitudes is determined by the transverse spin of the waveguide mode. The phase relation of the electric field components in the waveguide is determined by the spin-momentum locking phenomenon, and the transverse spin can be tuned by changing the parameters of the waveguide. By doing so, the authors demonstrate that the transverse spin angular momentum can be converted into OAM in the radiated vortex beam.

After reading this work, I completely agree with the authors that "This is the first demonstration of an SOI effect resulted from the interaction between the intrinsic OAM and transverse SAM of light" to the best of my knowledge. I believe this is a remarkable achievement. At first I was skeptical on how they would prove this to be the case, but I was very pleasantly surprised at the extremely clear argument that the authors use, particularly the elegant simplicity and clarity of Eq. (5) and the excellent geometric explanation of the transverse spin giving rise to a geometrical phase due to the angular rotation of the waveguide. More importantly, they perform a clear experimental demonstration that agrees extremely well with their result. I was completely convinced of their argument after seeing how "the relative intensities of the two dominant peaks, i.e., the two constituent CP vortices, can be changed by modifying σ ." On top of this, the authors even show the reciprocal situation, using the structure for unidirectional coupling of AM components in vortex beams.

In my opinion this work is a very important step in the field of angular momentum of light. It is

extremely interesting to the community, it has an excellent introduction, has very elegant theoretical foundations and simple explanations, is supported by very solid experiments, and should definitely be published in Nature Communications. With this very positive opinion, I have only minor suggestions for the authors:

- After Eq. (2) it is argued that σ is a real number because the electric fields are: but what about W_1 and W_2 ? are they real as well? In the supplementary information it is claimed that they are both real numbers, but I do not see any physical reason that they couldn't be complex in general. The polarizability of the scatterers might induce phase changes. The real character of W_1 and W_2 should be justified (at least in the supplementary materials).

- It would be much easier for readers if Eq. (3) was better introduced. The rotation matrix appears when converting the fields at each location from a cylindrical basis (azimuthal and radial electric field components, corresponding to longitudinal and transverse components of the waveguide) into the global rectangular (xy) coordinate system. This indeed introduces a geometric phase that depends on the transverse spin of the waveguide. This is very clear in the SI, but could be made much clearer in the main text.

- The notation used in page 9 with the angled brackets is not defined. I understand the authors are writing the spin and orbital angular momenta $\langle S, L \rangle$ but this should be clarified.

- Fig 2(d): Please add a label to the colormap: clarify it is showing $i E_{\text{long}}/E_{\text{trans}}$

- The introduction has a very complete review of the field. However, there are some works closely related to some aspects of the present work: the coupling of radiated light to and from transverse spin in the evanescent region near a rectangular integrated waveguide, also by using a laterally located scatterer, and also considering the different coupled polarization components for sorting and synthesis of different polarization states, similar to what is done here but with no OAM, in works [A,B,C,D] which the authors could consider including.

[A] ACS Photonics, vol. 1, pp. 762–767, 2014.

[B] Opt. Lett., vol. 39, pp. 1394–1397, 2014.

[C] Laser Photon. Rev., vol. 8, pp. L27–L31, 2014.

[D] Nano Lett., vol. 17, pp. 3139–3144, 2017.

- Finally, an optional suggestion. I believe the title doesn't make justice to the work: right now it seems a very general title and makes a reader wonder what is different to previous spin-orbit interactions based on spin-angular momentum. The title does not mention intrinsic OAM. Here for the first time they show interaction between intrinsic OAM and transverse SAM, the authors might consider mentioning IOAM in their title.

Reviewer #2 (Remarks to the Author):

This work investigates a spin-orbit interaction phenomenon of the coupling between the transverse spin of whispering-gallery (WG) modes and the intrinsic orbital angular momentum (OAM, vortex) of the outgoing far field.

The idea is interesting and novel, and it can be published in some form in Nature Communications, provided there is a clearly described theory and the corresponding experiment. Unfortunately, I found it to be very difficult to follow the explanations in the manuscript, both in its theoretical and experimental parts.

The mixture of various phases and topological numbers is confusing. It seems that p is the OAM quantum number of the WG mode, while q is the number of scatterers. Then, why the number of scatterers results in additional vortex phase $\exp(-iq\phi)$?

Does this assume a kind of diffraction-grating scattering along the azimuthal coordinate? What determines the momentum matching condition in this case?

Next, σ is defined as the ratio of the radial and azimuthal field components (which have a $\pi/2$ phase difference producing the transverse spin). Usually, in the literature on the topic,

σ denotes the wave helicity or spin, i.e., it is better to make the substitution

$\sigma \rightarrow 2\sigma/(1+\sigma^2)$ throughout the paper. This will simplify equations.

Then, as far as I understand, there should be a sort of selection rules for the coupling between different WG modes (characterized by p and q numbers) and different far-field vortex modes (characterized by the vortex charge ℓ and polarization σ). What are these selection rules?

It seems that they somehow correspond to the experimental results in Fig. 7 but this is not clearly explained.

The experimental part is also very difficult to understand. The WG modes are denoted as $WG_{\{m-n\}}$. What are the $\{m-n\}$ numbers? Where is the azimuthal index p ?

Why the full Stokes polarimetry is needed if only the transverse spin matters?

To summarize, I recommend the authors to considerably improve the presentation and to make the manuscript more reader-friendly and accessible for a general audience. There are too many technical details and plots but there is lack of clear explanation and emphasis for the central results of the work.

Reviewer 1:

[general comments omitted] – thanks for the very enthusiastic comments!

(1) After Eq. (2) it is argued that σ is a real number because the electric fields are: but what about W_1 and W_2 ? are they real as well? In the supplementary information it is claimed that

they are both real numbers, but I do not see any physical reason that they couldn't be complex in general. The polarizability of the scatterers might induce phase changes. The real character of W_1 and W_2 should be justified (at least in the supplementary materials).

Response: Thank you for the comment. This is a critical point indeed and should have been better clarified in the manuscript.

The two parameters W_1 and W_2 defined in this paper are used to characterize the scatterers' modulation on the field strength of electric components in evanescent wave scattering. The values of W_1 and W_2 (including being complex or real) are primarily subject to the type (i.e., Rayleigh or Mie) of scattering occurred, the geometric shape and material of the scatterers, as briefly discussed below.

- In this study, considering the constraints by fabrication techniques, square cuboidal scatterers are used for scattering the evanescent wave uniformly along the ring resonator waveguide. For scattering the predominantly transverse-polarized evanescent fields (i.e., E_r and E_ϕ as explained in the paper), the type of scattering only depends on the transverse cross-section (in the x-y plane) of particles. The square transverse cross-section of the scatterers in this work is a^2 ($a = 100$ nm), and

particle scattering of such cross-section at the given wavelength ($\lambda = 1590$ nm) can be well described by the Rayleigh approximation, as $a \ll \lambda$. As a result, the light scattered from each scatterer can be approximated by dipole radiation, with the phase locked to the driving transverse evanescent fields.

- The edges of the square cuboidal scatterers are strictly aligned with the local cylindrical coordinate axes (r, φ, z), so that the polarizability tensor of such particles is in fact a 3-by-3 diagonal matrix and the diagonal elements ($\alpha_{rr}, \alpha_{\varphi\varphi}, \alpha_{zz}$) represent the polarizabilities in the r, φ , and z directions, respectively. In other words, the scattering off such particles does not introduce crosstalk among orthogonal cylindrical field components.
- The polarizabilities $\alpha_{rr}, \alpha_{\varphi\varphi}$, and α_{zz} can be regarded as real numbers, because the absorption by SiN_x particles is negligible for 1590 nm light. Consequently, there is no instantaneous phase change in the Rayleigh scattering by the scatterers here, so that W_1 and W_2 can be assumed as real numbers.
- Interestingly, the assumption that W_1 and W_2 are real numbers has actually been justified by close examination of the measurements from the Stokes polarimetry performed on the near-field emission of devices (shown in the panel (v) curves of Figure 5), as the measured Stokes parameters as a function of the azimuthal position (φ) are all practically in-phase with the curves predicted by theory. In other words, any phase change induced by scattering of E_r and E_φ would have altered their phase difference from $\pm\pi/2$, so that tangible phase shifts between the measured and theoretical Stokes parameter curves would have appeared, which we didn't observe in experiment.

Revision 1: Following the Reviewer's advice, we have added a new note to the supplementary material, as the **Supplementary Note 1: Scattering Effect of Grating Elements**, to elaborate the scattering effect by such particles and the true character of W_1 and W_2 (including the analysis above). 1 figure, 4 reference papers, and 4 equations are included in this note, so that all other figures, references and equations in the supplementary material have been renumbered accordingly

Revision 2: In addition, the following revisions are also made to the main text and supplementary material to better clarify the discussion regarding W_1 and W_2 .

- The notations W_1 and W_2 have been changed to W_{rr} and $W_{\varphi\varphi}$, respectively, throughout the entire paper, for a clearer indication of their physical nature and connection to the polarizabilities of the particle.
- The following passage is inserted to the main text paragraph after Table 1 to justify our design of the scatterers.

“Intuitively, the simplest geometry of the scatterers for projecting the evanescent-field transverse-spin information into far-field with high fidelity is spherical Rayleigh particle²². However, the fabrication of sub-micron spherical scatterers, whether of metallic or dielectric nature, placed uniformly in the evanescent region along the resonator is rather challenging. Here we utilize square cuboid structures (as shown in Figure 1b), attached to the inner sidewall and of the same height as waveguide, as Rayleigh scatterers, allowing for both one-step etching in fabrication and highly uniform scattering of evanescent waves along the entire resonator (see Supplementary Note 1)”

(2) It would be much easier for readers if Eq. (3) was better introduced. The rotation matrix appears when converting the fields at each location from a cylindrical basis (azimuthal and radial electric field components, corresponding to longitudinal and transverse components of the waveguide) into the global rectangular (xy) coordinate system. This indeed introduces a geometric phase that depends on the transverse spin of the waveguide. This is very clear in the SI, but could be made much clearer in the main text.

Response: Thanks for the valuable comment.

Revision 1: Following your suggestion, **the description of Equation (3) and the local coordinate rotation it stands for has been clarified**, as the original remarks

“In addition, the vector fields of WGMs travelling along the resonator experience a rotation of local coordinate frame, which is described by the matrix”

are now extended to

“In addition, the vector fields of CCW WGMs travelling along the resonator experience a rotation of local coordinates (r, φ) with respect to the global reference frame (x, y) as $\varphi \cdot \mathbf{z}$ (see Figure 1c, and \mathbf{z} is unit vector), which can also be described by the matrix”. We believe this new description allows for easier understanding with a clearer reference to the illustration in Figure 1c.

Revision 2: In addition, in the introduction to the geometric phase induced by coordinate rotation (i.e., the passage before Equation (6)), a short reference to Equation (3) is now inserted: “... *a transverse-spin dependent geometric phase that stems from the rotation of local vector field, as represented by Equation (3)*”. However, we believe that the introduction and detailed explanation in the geometric phase is better given in the passage after Equation (5), as before, because the generalization of this geometric phase (given at the end of the passage after Equation (6)) involves the average spin angular momentum S_z , which is given in Equation (5). In this way, the discussion on the geometric phase appears more compact in the paper without redundancy.

(3) The notation used in page 9 with the angled brackets is not defined. I understand the authors are writing the spin and orbital angular momenta $\langle S, L \rangle$ but this should be clarified.

Response: Accepted with thanks.

Revision: Following your suggestion, a short sentence is now added after the first appearance of this notation (in the section of **Spin-orbit controlled unidirectional coupling**): “... i.e., $\langle 2\sigma/(1+\sigma^2), l_{TC}-2\sigma/(1+\sigma^2) \rangle$ and $\langle -2\sigma/(1+\sigma^2), l_{TC}+2\sigma/(1+\sigma^2) \rangle$. Here the notation $\langle s, l \rangle$ is used to indicate the SAM (s) and OAM (l) states of light.”

(4) Fig 2(d): Please add a label to the colormap: clarify it is showing $i E_{\text{long}}/E_{\text{trans}}$.

Response: Thanks for the advice.

Revision: A label as ‘ $iE_{\text{long}}/E_{\text{trans}}$ ’ is now added at the top center of Figure 2d.

(5) The introduction has a very complete review of the field. However, there are some works closely related to some aspects of the present work: the coupling of radiated light to and from transverse spin in the evanescent region near a rectangular integrated waveguide, also by using a laterally located scatterer, and also considering the different coupled polarization components for sorting and synthesis of different polarization states, similar to what is done here but with no OAM, in works [A,B,C,D] which the authors could consider including.

[A] ACS Photonics, vol. 1, pp. 762–767, 2014.

[B] Opt. Lett., vol. 39, pp. 1394–1397, 2014.

[C] Laser Photon. Rev., vol. 8, pp. L27–L31, 2014.

[D] Nano Lett., vol. 17, pp. 3139–3144, 2017.

Response: Thanks for the suggestion.

Revision: We agree that these works recommended by the Reviewer are highly related indeed to the subject of present work, especially the papers [A], [B], and [D]. These three papers are now included as the new references [37-39]. In addition, during the revision, we found two additional works that present excellent theoretical description to the effect of spin-direction locking in evanescent fields, so that these two papers are now referenced as [40, 41]. All the other references have been renumbered accordingly.

(6) Finally, an optional suggestion. I believe the title doesn't make justice to the work: right now it seems a very general title and makes a reader wonder what is different to previous spin-orbit interactions based on spin-angular momentum. The title does not mention intrinsic OAM. Here for the first time they show interaction between intrinsic OAM and transverse SAM, the authors might consider mentioning IOAM in their title.

Response: Thanks for the inspiring advice.

We have tried to take the Reviewer's suggestion and considered new titles such as 'Interaction between intrinsic orbit angular momentum and engineered transverse spin angular momentum'. But titles like this explicitly highlighting the 'intrinsic orbital angular momentum' and 'transverse spin angular momentum' will inevitably look long-winded to some extent, because of the repeated appearance of 'angular momentum'.

On the other hand, the term 'spin-orbit interaction' we use in the current title is a clear and concise phrase that summarizes the general phenomena that involve the spin and orbital angular momentum of light. We feel that this widely known concept conveys the very central idea of this paper to readers in a straightforward and efficient manner. Although the intrinsic nature of orbital angular momentum studied in this work is not specified in the title, we have clearly introduced in the first sentence of Abstract the two objects being investigated: "*We report the first demonstration of a direct interaction between the extraordinary transverse spin angular momentum in evanescent waves and the intrinsic orbital angular momentum in optical vortex beams.*" Therefore, we decided to stick to the current title.

Reviewer 2:

This work investigates a spin-orbit interaction phenomenon of the coupling between the transverse spin of whispering-gallery (WG) modes and the intrinsic orbital angular momentum (OAM, vortex) of the outgoing far field.

The idea is interesting and novel, and it can be published in some form in Nature Communications, provided there is a clearly described theory and the corresponding experiment.

Unfortunately, I found it to be very difficult to follow the explanations in the manuscript, both in its theoretical and experimental parts.:

(1) The mixture of various phases and topological numbers is confusing. It seems that p is the OAM quantum number of the WG mode, while q is the number of scatterers. Then, why the number of scatterers results in additional vortex phase $\exp(-iq\phi)$?

Does this assume a kind of diffraction-grating scattering along the azimuthal coordinate? What determines the momentum matching condition in this case?

Response: Thank you for the comment. The micro-ring resonator based device indeed is embedded with an 2^{nd} -order angular diffraction grating to diffract the bounded WGMs out as propagating vortex modes. We first reported such a device as a vortex beam emitter several years ago, and the fundamentals and principles of the device have been elaborated in our previous papers, e.g., ref. [46] in the manuscript. However, with this comment from the Reviewer, we realize that giving a brief introduction of the working principle of the structure (at least in the supplementary notes) will be very helpful for the understanding of a broad audience.

Revision 1: We insert a new independent note (including a schematic diagram) in the supplementary material to explain the interaction between the WGMs, grating and emitted vortex beams, i.e., the angular phase matching (or angular momentum conservation) condition. This note is now referred to as **Supplementary Note 2. Phase-matching in CVV Scattering**, in which the derivation of the additional vortex phase $\exp(-iq\phi)$ is also presented. Note that all the equations, figures and references have been renumbered accordingly.

Revision 2: In addition, some more explanation of the diffraction phase $\exp(-iq\phi)$ specifically is given in the passage below Equation (8) of the current **Supplementary Note. 3**. The relevant remarks has been modified from (in the previous Supplementary Note. 1)

“ $\delta(\varphi) = \mp q\varphi$ is the phase imparted on the first-order diffracted wave derived using the coupled-mode theory (cf. supplementary material of ref. [1]) and q the number of grating elements” to “Note that the phase acquired in scattering $\delta(\varphi)$ is not an instantaneous phase shift to the scattered fields, but only represents the relative phase delay between the scattered waves at different locations. From the phase-matching condition above (i.e., $l = p - q$) and the spatial phase of WGMs (i.e., $\pm p\varphi$), it’s straightforward to find that $\delta(\varphi) = \mp q\varphi$. This result agrees with the phase imparted on the first-order wave in 2nd-order-grating diffraction derived using coupled-mode theory (cf. supplementary material of ref. [4])”.

Revision 3: Accordingly, the introduction to the phase $\delta(\varphi)$ in the main text has also been changed from “ $\delta(\varphi) = -q\varphi$ (see supplementary material of ref. 41) is the azimuthal phase acquired by the second-order grating scattering” to “ $\delta(\varphi) = -q\varphi$ (Supplementary Note 2 and 3) is the azimuthal phase acquired by the second-order grating scattering”.

(2) Next, σ is defined as the ratio of the radial and azimuthal field components (which have a $\pi/2$ phase difference producing the transverse spin). Usually, in the literature on the topic, σ denotes the wave helicity or spin, i.e., it is better to make the substitution $\sigma \rightarrow 2\sigma/(1+\sigma^2)$ throughout the paper. This will simplify equations.

Response: Thank you very much for the comment. We agree that σ is usually used to denote the helicity or spin state of many text-book light beams, such as plane wave and Gaussian beam. But for complex light fields of inhomogeneous polarization state across the transverse plane or of high non-paraxiality, which means the polarization state is a function of transverse position (i.e., $\sigma \sim \sigma(x, y)$), generally the spin angular momentum (SAM) flux across the transverse plane (or, the SAM per unit longitudinal length per energy, S_z as used in this paper) is used instead to represent the average SAM or spin property of the beam. Therefore, using a single averaged helicity σ in this case to characterize the whole light beam is not very meaningful.

Meanwhile, the quantity of $2\sigma/(1+\sigma^2)$ that appears multiple times in the paper actually stands for the spatial phase gradient along the azimuthal direction (as shown in Equation (S17) of the new Supplementary Note. 5) resulting from the rotation of evanescent waves along the resonator. That is, **$2\sigma/(1+\sigma^2)$ has a completely different physical nature to σ** , and therefore the substitution suggested by the Reviewer is inappropriate.

On the other hand, we do have given it a serious consideration about defining $2\sigma/(1+\sigma^2)$ as a new quantity that resembles a topological charge, similar to l_{TC} in the paper, and denotes the change rate of spatial phase solely contributed by the rotation of transverse spin. However, just as suggested by the Reviewer, the amount of physical quantities involved in this paper might already be a bit overwhelming for the general audience and we should be extremely careful in introducing new concepts. In addition, we believe that presenting the equations (e.g., Equations (5) and (6)) with the part of $2\sigma/(1+\sigma^2)$ unsubstituted can be even beneficial by emphasizing the effect of the transverse spin state σ in these equations and the spin-orbit interaction phenomenon they stand for. Considering all the factors above, we decide not to use any substitution for $2\sigma/(1+\sigma^2)$.

(3) Then, as far as I understand, there should be a sort of selection rules for the coupling between different WG modes (characterized by p and q numbers) and different far-field vortex modes (characterized by the vortex charge ℓ and polarization σ). What are these selection rules? It seems that they somehow correspond to the experimental results in Fig. 7 but this is not clearly explained.

Response: Thanks for the comment. Indeed, the purpose of the incident vortex experiment is to further establish the selection rules that have been shown in the emission process. This selection rule is now provided and included in the supplementary material (i.e., **Supplementary Note 2. Phase-matching in CVV Scattering**, as shown above in the response to the first comment of Reviewer 2). In this new Note, the interaction among the whispering-gallery modes, grating, and the scattered vortex modes, as well as the selection rule of CVV emission, is described with a simple phase-matching condition (i.e., Equation (S5) in Supplementary Note. 2).

Revision 1: Nevertheless, this selection rule of mode coupling into devices is now also reiterated in the section of **Spin-orbit controlled unidirectional coupling** of the main text, shown in the first paragraph as “... *Additionally, the excited modes must follow the phase-matching condition for the resonance as WGMs (Supplementary Note 2), which can be written here as $l_{in} + \sigma_{in} = l_{res}$ and l_{in} is the OAM state of incident vortex*”.

Revision 2: In addition, following the Reviewer’s suggestion, in order to improve the presentation of the results in Figure 7 so that it is easier to understand by readers, the **entire section of Spin-orbit controlled unidirectional coupling has been literally rewritten**, and the key revisions made include

- **Figure 7 is now replotted.**
 - A new schematic plot showing the selection rule of CVV reception is now provided as Figure 7a.
 - The previous bar graphs in Figure 7 showing the received power at the two access ports have now been replaced with three new 3-D bar graphs, in order to present the results and details more clearly.
 - The results are also re-organized, and now Figures 7b-7d respectively show the results with incidence of SAM states $\sigma_{in} = +1, 0, -1$. In this way, the spin-direction locking effect is more evident, as only bars of a single color are predominant if the incident light is of circular polarizations. Moreover, the spin-orbit jointly controlled unidirectional coupling is presented more clearly, as now the bar graphs show the identity-matrix-like distribution in a straightforward accordance with the selection rule.
 - The caption of Figure 7 is revised accordingly.
- **Table 2 in the previous version is removed**, as it may cause some confusion to readers and also appear rather redundant with the new Figure 7.
- The text of whole section is substantially revised to better explain the selection rule and the new Figure 7.

(4) The experimental part is also very difficult to understand. The WG modes are denoted as $WG_{\{m-n\}}$. What are the $\{m-n\}$ numbers? Where is the azimuthal index p ?

Response: Thanks for the comment. The notation WG_{w-h} in the paper is used to refer to the sample devices of various ring-resonator waveguide (WG) dimensions, and the subscripts w and h denote the width and height (in the unit of 100-nm) of the rectangular waveguides, respectively.

Revision 1: To better clarify this notation in the paper, the following sentence is added to the paragraph before Table 1:

“... where the subscripts w and h in the notation for waveguide sample WG_{w-h} indicate the width and height of waveguide (in the unit of 100-nm), respectively”.

(5) Why the full Stokes polarimetry is needed if only the transverse spin matters?

Response: The full Stokes polarimetry is needed to completely characterize the spatial variation of the polarization state because it reveals important information about how the

near-field transverse SAM is projected onto the CVV's longitudinal spin. In particular the cyclic oscillation of the Stokes parameters reveals the existence of the geometric phase and also provides clues to the scattering process (as discussed per Reviewer 1's comment on the real or complex nature of the coefficients $W_{1,2}$).

Revision 1: To better introduce the Stokes polarimetry to the context, we revised the following sentence “*Secondly, Stokes polarimetry is performed to characterize the local transverse-spin state distribution in near-field CVVs*” into “*The results above also indicate that the transverse spin state once bounded at near-field now manifests itself as the longitudinal (i.e., along z direction) spin state in the propagating CVVs. Therefore, by performing the spatially resolved Stokes polarimetry to the near-field image of CVVs, the local transverse-spin state distribution in near-field CVVs can be revealed (see Supplementary Note 4)*”.

Revision 2: Moreover, to provide more detailed explanation on the results of Stokes polarimetry, the discussion on the results shown in Figure 5 is rewritten from the previous text of “*The agreement between the theoretical curves and measured dots shown in the S_3 plots of (v) validates the overall effect of waveguide geometry on the transverse-spin state in evanescent waves. For devices of larger $|\sigma|$, e.g., WG_{6-8} , S_1 and S_2 oscillate less, indicating local polarization states of larger ellipticity*” into “*First, the dots of each measured S_1 or S_2 in (v) complete the oscillation of two full sinusoidal cycles around the resonator, following the theory predictions (curves). And thus, the rotation of near-field transverse spin state shown in Figure 1c is visualized, thereby indicating the existence of the geometric phase predicted in Equation (6). On the other hand, the modulation effect of waveguide geometry on the local transverse-spin state in evanescent waves can be validated, by comparing the amplitude of S_1 (S_2) oscillation from the four devices in Figure 5. Sample WG_{6-8} , with which S_1 and S_2 hardly oscillate, produces the largest local polarization ellipticity of the four in the evanescent region. More discussion on the results of Stokes polarimetry is provided in Supplementary Note 8.*”

Revision 3: In addition, the passage shown below is now removed from the main text, as these detailed discussions may cause distraction to readers from the most important results. “~~*Generally, the jitters in the measured results are attributed to the non-uniformity of fabricated gratings, as well as the decaying intensity of WGMs along the resonator. The deviation of measurements from the theory is more evident with devices of smaller $|S_3|$. This is possibly caused by the light that is scattered from the other (outer) side of waveguide,*~~

~~carrying the opposite σ , due to sidewall roughness. In some devices, standing-wave-like patterns (e.g., map (iv) in Figure 5c) are introduced by the interference of scattered TE and TM modes, because in these waveguide designs these two modes are more degenerate and single polarization mode excitation is more critical to polarization control in mode launching~~'. Instead, these are moved to the supplementary material now as the new **Supplementary Note 8**.

(6) To summarize, I recommend the authors to considerably improve the presentation and to make the manuscript more reader-friendly and accessible for a general audience. There are too many technical details and plots but there is lack of clear explanation and emphasis for the central results of the work.

Response: Many thanks for the advice. Following this suggestion, we have made substantial revisions to improve the presentation of our results. The revisions regarding improving the presentation that have been mentioned above are here briefly summarized:

- The **physical nature of scatterers and their effect on evanescent waves** are more explicitly explained, and the new Supplementary Note 1 is added.
- The **fundamental physics of this device in vortex beam emission and receiving** are more clearly explained, including the selection rules (phase-matching condition) of mode coupling (Supplementary Note 2 is added) and the physical nature of coordinate rotation shown in Equation (3).
- The **discussion on the Stokes polarimetry** is significantly expanded and more explanation on the results is provided.
- The entire section of **Spin-orbit controlled unidirectional coupling** is rewritten to better explain the results shown in Figure 7, including a schematic plot added in Figure 7 and Figure 7 results re-organized and re-plotted.

Apart from the revisions mentioned above, **other revisions and clarifications we made to provide more reader-friendly presentation include**

In the introductory passages before Results

- The word 'incident' is inserted into the sentence "... we further demonstrate directional coupling of optical vortices into this integrated photonic circuitry, with the direction of the waveguide modes jointly controlled by the incident spin and orbital AM states ..."

In the section of Interaction of transverse-spin and OAM

- A reference to the supplementary material is added to the first sentence: “*The emission of CVVs from such structures can be generally described in the form of transfer matrices as $E_{out} = M_2 \cdot M_1 \cdot E_{in}$ (see Supplementary Note 3 for details)*”.
- The words ‘evanescent wave’ are removed from the sentence “*Assuming the WGM ~~evanescent wave~~ maintains a uniform distribution around the resonator*” for simpler expression.
- The notations for the radial and azimuthal field components are now added in the following sentence “... *the generic input light for the matrices is the inner sidewall evanescent wave and can be written in the locally transverse (E_r) and longitudinal (E_ϕ) polarization basis*” for clearer indication.
- In the first sentence after Equation (2), the discussion on the transverse spin state is revised from “... *and it directly characterizes the (spatial) transverse-spin density in the evanescent wave as $S_\perp \propto \sigma$ (refs 6,11)*” to “*and it directly characterizes the spatial density of transverse spin^{6,11}*”, as here using the additional notation S_\perp for transverse spin density may only introduce confusion for readers. Accordingly, the notation S_\perp has been removed throughout the whole paper.
- The sentence after Equation (5) “*Note that S_z here, which should be distinguished from the spatial transverse spin density S_\perp , is the SAM in CVVs averaged over the transverse x-y plane*” is now revised as “*Note that S_z is the SAM in the emitted CVVs averaged over the transverse x-y plane, which should be distinguished from the spatial density of transverse spin in evanescent waves*”.
- The sentence before Equation (6) is revised with the insertion of “... *the Pancharatnam phase^{49,52}, which is extensively used for comparing the phase between light fields of different polarization states, is here used to describe*” to introduce and justify the involvement of Pancharatnam phase.
- ‘CCW’ is added to the sentence “... *has a pure geometric nature and arises from the rotation of local transverse-spin state while CCW WGMs travel around the resonator*” for more accurate description.

In the section of Transverse spin engineering

- The following passage is added when introducing the device parameters to justify our design of the scatterers’ geometry (as discussed in the response to the first question of Reviewer 1)

“Intuitively, the simplest geometry of the scatterers for projecting the evanescent-field transverse-spin information into far-field with high fidelity is spherical Rayleigh particle²². However, the fabrication of sub-micron spherical scatterers, whether of metallic or dielectric nature, placed uniformly in the evanescent region along the resonator is rather challenging. Here we utilize square cuboid structures (as shown in Figure 1b), attached to the inner sidewall and of the same height as waveguide, as Rayleigh scatterers, allowing for both one-step etching in fabrication and highly uniform scattering of evanescent waves along the entire resonator (see Supplementary Note 1)”.

- As a result, the description of the device and scatterers is revised from *“For the 8 sample devices, the ring radius of 80 μm is used. For each device, $q = 517$ scatterers are embedded on the inner-sidewall of ring. The ratio of evanescent cylindrical components may be perturbed by the presence of scatterers in the evanescent region, as represented by matrix \mathbf{M}_1 . In this proof-of-principle study, we consider square-shape scatterers protruding from the waveguide sidewall. Each scatterer has the constant area of 100 nm by 100 nm, but is in the same height as the ring waveguide”* to *“For the 8 sample devices, the ring radius $R = 80 \mu\text{m}$. For each device, the number of scatterers $q = 517$, and each scatterer has a planar size 100 nm by 100 nm”.*
- In addition, a new Supplementary Note is added as Note 1 to explain the scattering effect of the grating elements with the fundamental Rayleigh scattering theory (as shown in the response to the first question of Reviewer 1).

In the section of Polarization and transverse-spin state characterization

- The word ‘global’ is added to the sentence “... and therefore here the components E_r and E_ϕ are measured to characterize the average global ellipticity in the cylindrical basis ...” to better distinguish the results shown in Figure 4 and those in Figure 5.
- Also, the word ‘local’ is used in “... by performing the spatially resolved Stokes polarimetry to the near-field image of CVVs, the local transverse-spin state distribution in near-field CVVs can be revealed ...” for the same reason.

In the section of Spin-orbit controlled unidirectional coupling

- In Table 2, the topological charges, as well as the SAM and OAM charges, are now marked in blue (red), if in the column of access port 1 (2). This difference in color for these two access ports is in accordance with the colors of bars shown in Figure 7 representing the power received at the two ports, so that it will be more intuitive for readers to understand the results in Figure 7.

In the Supplementary Note 3

- Some remarks are inserted at the beginning to conclude the phase-matching condition presented in Note 2 and to introduce the formulation in Note 3: *“The phase-matching condition that rules the WGM and grating scatterers has been discussed in the previous Note to show the essence of this device. However, to fully present the interaction between the transverse spin in evanescent waves and the OAM in CVVs, a more explicit formulation is presented here”*.

Additional revisions and corrections:

Abstract

- Typo corrected by removing the ‘in’: *“By tapping the evanescent wave of ~~in~~ a whispering-gallery-mode-based optical vortex emitter and engineering the transverse-spin state carried therein”*.
- Missing preposition ‘of’: *“This unconventional interplay between the spin and orbital angular momenta allows the regulation of the spin-orbital angular momentum states ...”*.
- The last sentence is simplified: *“... and can enable a variety of functionalities ~~employing spin and orbital angular momenta of light~~ in applications such as communications and quantum information processing”*.

Figure 1

- The caption is simplified to reduce redundancy: *“(a) Schematic of the platform for the investigation of transverse spin induced SOI effect. ~~A single-transverse-mode ring resonator is coupled with an access waveguide and embedded with sub-wavelength scatterers arranged as 2nd-order grating in the evanescent-wave region.~~ (b) ~~Each WGM possesses transverse spin of opposite signs in the inner- and outer-~~*

~~resonator evanescent waves, and~~The clock-wise (CW) and counter clock-wise (CCW) WGMs present opposite transverse spins on each side of the resonator ...”

Figure 2

- The error in the caption is corrected: “... and the dashed rectangular indicates the waveguide of 0.6 μm width and 0.8 μm height” is revised to “... and the dashed rectangular indicates the waveguide of 0.8 μm width and 0.6 μm height”.

In the section of Transverse spin engineering

- The inappropriate adverb ‘effectively’ is removed: “More importantly, a contour map of this ratio is plotted in Figure 2d, in which an ~~effectively~~ variable ratio of the two components can be observed over various waveguide dimensions”.
- ‘refractive index’ is acronymized as ‘RI’ in the last sentence before Table 1, and the refractive index of silicon is mentioned for comparison: “ SiN_x waveguide is employed for its moderate refractive index (RI ~ 2.01) so that a larger range of transverse-spin state can be accessed than other materials (e.g., silicon with RI ~ 3.5)”.

Table 1

- The following annotation is removed due to redundancy: “~~*The ring radius of all sample devices is 80 μm , gap between ring and access waveguide is 200 nm, and each square shape scatterer is 100 nm by 100nm (with the same height as waveguide)~~”.

In the section of Polarization and transverse-spin state characterization

- An error is corrected in the following sentence by replacing ‘ $|\sigma|$ ’ with ‘ ε ’:
“Particularly, near-CP ($\varepsilon \approx 1$) CVVs are observed with devices WG_{4-10} and WG_{6-8} , indicating that the reduced superposition of single spin-orbital eigen-state vortices predicted by Equation (4) can be reached ...”.

In Methods

- The methods for Numerical simulations is now revised as “Numerical simulations. Numerical simulations are performed with the finite difference eigenmode solver (FDE, Lumerical Solutions, Inc.)”, because the calculation of W_1 and W_2 previously shown here is now moved into Supplementary Note 1.

- The description of ‘Experimental setups’ is revised with more details added in the parentheses: “*Experimental setups for device characterizations (including polarization ellipticity and Stokes polarimetry), SOI measurement and spin-orbit controlled unidirectional coupling are shown and explained in Supplementary Note 6 and 9*”.

In supplementary material

- All equations and references are now referred to with an additional ‘S’ before their number, e.g., Equation (S1) and ref. [S1], to be better distinguished from those in the main text.

Reviewers' Comments:

Reviewer #1 (Remarks to the Author):

See document attached.

Reviewer #2 (Remarks to the Author):

The authors have addressed the referees' comments and have considerably improved the presentation in the paper. I still think that notation $2\sigma/(1+\sigma^2)$ should be changed to σ . This is just a notation, and it will considerably simplify the equations. Moreover, the fact that this quantity is the azimuthal phase gradient exactly emphasizes that it is the relevant spin quantity, see PRL 101, 030404 (2008). This is just a minor remark; otherwise, the paper can be published in Nature Communications.

Comments from Reviewer #1:

The authors successfully answered all my concerns about the paper. Indeed the real character of W_{rr} and $W_{\phi\phi}$ is now clear. The absorption cross section of a dipolar scatterer is proportional to the imaginary part of the polarizability. Therefore if the scatterers are not absorptive, the polarizability can be said to be real.

The authors' improved description of the experimental results is very satisfactory, and I think they made an excellent work in summarizing their results for the inverse experiments in such a short space.

I have the following comments on the resubmitted version:

A- The discussion of the transverse spin in the evanescent region of the waveguides, together with the figures of longitudinal and transverse field, reminded me very strongly of a paper "Espinosa-Soria, A. & Martinez, A. Transverse Spin and Spin-Orbit Coupling in Silicon Waveguides. IEEE Photonics Technol. Lett. 28, 1561–1564 (2016)" which could be cited due to the similarities.

B- I would like to convince the authors that reviewer 2 point 2 was actually a very valid point that I strongly suggest they follow. I feel that the authors did not completely understand the reviewers comment and their answer missed a key point, in my opinion. I explain in detail below:

I completely agree with the reviewer 2 that some confusion is added by the notation of σ used in Eqs. (2,5,6). This notation is typically reserved for the spin of the field, and actually the authors are using it for a quantity which is very similar to the local spin of the field (it is identical in the cases -1, 0 and 1), which adds to the confusion. In fact the authors describe σ as "characterising the transverse spin", and later, σ is directly compared on equal footing with the SAM S_z , e.g. "the variation in magnitude from the local density (σ) to the average SAM (S_z) is associated with [...]". The use of the symbol and the descriptions in the text make a reader continually think that σ is the usual definition for the local density of transverse spin, but it is not. Even Fig 1(b,c) shows arrows representing the circular polarization of the transverse spin in the evanescent wave, next to the symbol σ , reinforcing this interpretation. In fact, later experimental discussions of illuminating vortex beams use σ as the spin of the illumination. Overall, I think that the choice of measurement parameter σ was unfortunate.

I can find two ways of solving this:

(1) Rename σ to some other symbol such as κ (I will use κ from now on in this document to refer to the author's σ)

$$\kappa = \begin{cases} -i(E_{rr}/E_{\phi\phi}) & |E_{rr}| < |E_{\phi\phi}| \\ +i(E_{\phi\phi}/E_{rr}) & |E_{rr}| > |E_{\phi\phi}| \end{cases}$$

Note that this definition has some drawbacks, namely the conditional definition, and also the fact that it inherently assumes that $E_{\phi\phi}$ is in quadrature phase to E_{rr} in order for κ to be real. This is not as elegant as it could be.

(2) Use the correct usual definition for the local spin of the evanescent wave and re-write the equations in terms of the proper definition of σ . This is what reviewer 2 suggested and I agree with him. I strongly suggest the authors to perform this substitution because it simplifies all the expressions in the paper and helps with the intuitive understanding.

The usual definition for the local spin is the following (which I will call σ and I will compare with the author's κ):

$$\sigma = \frac{|E^+|^2 - |E^-|^2}{|E^+|^2 + |E^-|^2}$$

This is a very natural and easy to understand definition of local spin, where E^+ and E^- are the right and left handed components of the polarization vector in the coordinate basis (r, ϕ) , obtained as the projection of the Jones vector $\mathbf{E} = (E_{rr}, E_{\phi\phi})$ into the unit vectors σ^+ and σ^- for local circular polarizations in the (r, ϕ) basis.

$$E^+ = \mathbf{E} \cdot (\sigma^+)^* = \begin{pmatrix} E_{rr} \\ E_{\phi\phi} \end{pmatrix} \cdot \frac{1}{\sqrt{2}} \begin{pmatrix} 1 \\ -i \end{pmatrix}^*$$

$$E^- = \mathbf{E} \cdot (\sigma^-)^* = \begin{pmatrix} E_{rr} \\ E_{\phi\phi} \end{pmatrix} \cdot \frac{1}{\sqrt{2}} \begin{pmatrix} 1 \\ i \end{pmatrix}^*$$

Using this substitution one can write:

$$\sigma = \frac{|E_{rr} + iE_{\phi\phi}|^2 - |E_{rr} - iE_{\phi\phi}|^2}{|E_{rr} + iE_{\phi\phi}|^2 + |E_{rr} - iE_{\phi\phi}|^2}$$

If we compare the values of κ (the author's choice for characterising the transverse spin) and σ (the usual choice) we find that they are very similar, they both "characterise the transverse spin" because the cases +1 and -1 correspond to RCP and LCP pure circular polarizations, while the case 0 corresponds to linear polarizations. However, the values of σ and κ differ for all intermediate elliptical polarization cases.

As an example, let's take the following normalized Jones vector for the local polarization, varying the ratio of the components in polar basis, but with constant phase difference always in quadrature:

$$\begin{pmatrix} E_{rr} \\ E_{\phi\phi} \end{pmatrix} = \begin{pmatrix} \cos \alpha \\ i \sin \alpha \end{pmatrix}$$

So varying α we can "tune" the polarization between circular and linear through all the elliptical cases. If we plot the values of κ and σ as a function of α one gets:

Note that, if we assume that $E_{\phi\phi}$ is in quadrature phase with E_{rr} as in the figure above, then it can be shown analytically that $\sigma = \frac{2\kappa}{1+\kappa^2}$, which is a factor that appears throughout several equations in the paper, and which could be simplified to show only σ .

In fact, it is well known that the geometric phase acquired due to rotations of local coordinates by an angle ϕ can be easily written in terms of the local spin σ of the polarization as the simple product $\sigma\phi$. **See for example Box 2 in the review article [11]**. This is a known result that will yield a very elegant equation $\Phi_P = \ell_{TC}\phi + \sigma\phi$ instead of the complex-looking Eqs. (6, S17, S18). It is a much more elegant explanation for the geometric phase term than all the calculations involving matrix M2 and the supplementary note 5.

This linear relation between geometric phase and spin is the reason that the authors thought that “the quantity of $2\sigma/(1+\sigma^2)$ that appears multiple times in the paper actually stands for the spatial phase gradient along the azimuthal direction resulting from the rotation of evanescent waves along the resonator” as stated in the rebuttal letter. The authors then claim “That is, $2\sigma/(1+\sigma^2)$ has a completely different physical nature to σ , and therefore the substitution suggested by the Reviewer is inappropriate”. I think that the authors missed the key fact that the local spin is indeed exactly equal to the spatial phase gradient due to the rotation of coordinates.

I hope this has convinced the authors that the use of the proper definition of σ will increase the readability of their work, and facilitate greatly the intuitive descriptions. I see no real advantages in the use of the measure κ .

Overall I think the quality of this work is excellent and recommend it for publication after the revisions above are considered.

Response to Reviewer's Comments on Nature Communications Manuscript NCOMMS-17-22490A:

'Spin-orbit interaction of light induced by transverse spin angular momentum engineering' by Zengkai Shao *et al.*

Reviewer 1:

The authors successfully answered all my concerns about the paper. Indeed the real character of W_{rr} and W_{pp} is now clear. The absorption cross section of a dipolar scatterer is proportional to the imaginary part of the polarizability. Therefore if the scatterers are not absorptive, the polarizability can be said to be real. The authors' improved description of the experimental results is very satisfactory, and I think they made an excellent work in summarizing their results for the inverse experiments in such a short space.

I have the following comments on the resubmitted version:

(1) The discussion of the transverse spin in the evanescent region of the waveguides, together with the figures of longitudinal and transverse field, reminded me very strongly of a paper "Espinosa-Soria, A. & Martinez, A. Transverse Spin and Spin-Orbit Coupling in Silicon Waveguides. IEEE Photonics Technol. Lett. 28, 1561–1564 (2016)" which could be cited due to the similarities.

Response: Thank you for the comment. This paper is now added to the manuscript as the new reference [42]. Other references have been re-numbered accordingly.

(2) I would like to convince the authors that reviewer 2 point 2 was actually a very valid point that I strongly suggest they follow. I feel that the authors did not completely understand the reviewers comment and their answer missed a key point, in my opinion. I explain in detail below:

I completely agree with the reviewer 2 that some confusion is added by the notation of σ used in Eqs. (2,5,6). This notation is typically reserved for the spin of the field, and actually the authors are using it for a quantity which is very similar to the local spin of the field (it is identical in the cases $\pm 1, 0$ and 1), which adds to the confusion. In fact the authors describe σ as "characterising the transverse spin", and later, σ is directly compared on equal footing with the SAM S_z , e.g. "the variation in magnitude from the local density (σ) to the average SAM (S_z) is associated with [...]". The use of the symbol and the descriptions in the text make a reader continually think that σ is the usual definition for the local density of transverse spin, but it is not. Even Fig 1(b,c) shows arrows representing the circular polarization of the

transverse spin in the evanescent wave, next to the symbol σ , reinforcing this interpretation. In fact, later experimental discussions of illuminating vortex beams use σ as the spin of the illumination. Overall, I think that the choice of measurement parameter σ was unfortunate.

I can find two ways of solving this:

(i) Rename σ to some other symbol such as κ (I will use κ from now on in this document to refer to the author's σ)

$$\kappa = \begin{cases} -i(E_{rr}/E_{\varphi\varphi}) |E_{rr}| < |E_{\varphi\varphi}| \\ +i(E_{\varphi\varphi}/E_{rr}) |E_{rr}| > |E_{\varphi\varphi}| \end{cases}$$

Note that this definition has some drawbacks, namely the conditional definition, and also the fact that it inherently assumes that $E_{\varphi\varphi}$ is in quadrature phase to E_{rr} in order for κ to be real. This is not as elegant as it could be.

(ii) Use the correct usual definition for the local spin of the evanescent wave and re-write the equations in terms of the proper definition of σ . This is what reviewer 2 suggested and I agree with him. I strongly suggest the authors to perform this substitution because it simplifies all the expressions in the paper and helps with the intuitive understanding.

The usual definition for the local spin is the following (which I will call σ and I will compare with the author's κ):

$$\sigma = \frac{|E^+|^2 - |E^-|^2}{|E^+|^2 + |E^-|^2}$$

This is a very natural and easy to understand definition of local spin, where E^+ and E^- are the right and left handed components of the polarization vector in the coordinate basis (r, φ) , obtained as the projection of the Jones vector $\mathbf{E} = (E_{rr}, E_{\varphi\varphi})$ into the unit vectors $\boldsymbol{\sigma}^+$ and $\boldsymbol{\sigma}^-$ for local circular polarizations in the (r, φ) basis.

$$E^+ = \mathbf{E} \cdot (\boldsymbol{\sigma}^+)^* = \begin{pmatrix} E_{rr} \\ E_{\varphi\varphi} \end{pmatrix} \cdot \frac{1}{\sqrt{2}} \begin{pmatrix} 1 \\ -i \end{pmatrix}^*$$

$$E^- = \mathbf{E} \cdot (\boldsymbol{\sigma}^-)^* = \begin{pmatrix} E_{rr} \\ E_{\varphi\varphi} \end{pmatrix} \cdot \frac{1}{\sqrt{2}} \begin{pmatrix} 1 \\ i \end{pmatrix}^*$$

Using this substitution one can write:

$$\sigma = \frac{|E_{rr} + iE_{\varphi\varphi}|^2 - |E_{rr} - iE_{\varphi\varphi}|^2}{|E_{rr} + iE_{\varphi\varphi}|^2 + |E_{rr} - iE_{\varphi\varphi}|^2}$$

If we compare the values of κ (the author's choice for characterising the transverse spin) and the σ (the usual choice) we find that they are very similar, they both "characterise the transverse spin" because the cases $+1$ and -1 correspond to RCP and LCP pure circular

polarizations, while the case 0 corresponds to linear polarizations. However, the values of σ and κ differ for all intermediate elliptical polarization cases.

As an example, let's take the following normalized Jones vector for the local polarization, varying the ratio of the components in polar basis, but with constant phase difference always in quadrature:

$$\begin{pmatrix} E_{rr} \\ E_{\phi\phi} \end{pmatrix} = \begin{pmatrix} \cos \alpha \\ i \sin \alpha \end{pmatrix}$$

So varying α we can “tune” the polarization between circular and linear through all the elliptical cases. If we plot the values of κ and σ as a function of α one gets:

Note that, if we assume that $E_{\phi\phi}$ is in quadrature phase with E_{rr} as in the figure above, then it can be shown analytically that $\sigma = \frac{2\kappa}{1+\kappa^2}$, which is a factor that appears throughout several equations in the paper, and which could be simplified to show only σ .

In fact, it is well known that the geometric phase acquired due to rotations of local coordinates by an angle φ can be easily written in terms of the local spin σ of the polarization as the simple product $\sigma\varphi$. **See for example Box 2 in the review article [11]**. This is a known result that will yield a very elegant equation $\Phi_P = \ell_{TC}\varphi + \sigma\varphi$ instead of the complex looking Eqs. (6, S17, S18). It is a much more elegant explanation for the geometric phase term than all the calculations involving matrix M2 and the supplementary note 5.

This linear relation between geometric phase and spin is the reason that the authors thought that “the quantity of $2\sigma/(1+\sigma^2)$ that appears multiple times in the paper actually stands for the spatial phase gradient along the azimuthal direction resulting from the rotation of evanescent waves along the resonator” as stated in the rebuttal letter. The authors then claim “That is,

$2\sigma/(1+\sigma^2)$ has a completely different physical nature to σ , and therefore the substitution suggested by the Reviewer is inappropriate". I think that the authors missed the key fact that the local spin is indeed exactly equal to the spatial phase gradient due to the rotation of coordinates.

I hope this has convinced the authors that the use of the proper definition of σ will increase the

readability of their work, and facilitate greatly the intuitive descriptions. I see no real advantages in the use of the measure κ .

Overall I think the quality of this work is excellent and recommend it for publication after the revisions above are considered.

Response: Many thanks for the enlightening comments! Now we fully understand the two Reviewers' concern, and agree that the second definition suggested above is a better choice for the transverse spin definition. We have changed the definition of the transverse spin state throughout the entire paper, and in the meantime carefully revised the relevant descriptions and discussions as follows.

Revision 1: First, new notations E_{rr} and $E_{\varphi\varphi}$ are now used (**in the passage before Equation (2)**) to represent the evanescent field components after the modulation of grating perturbation (i.e., $E_{rr} = W_{rr}E_r$ and $E_{\varphi\varphi} = W_{\varphi\varphi}E_\varphi$), such that the explanation of grating-light interaction is further simplified.

Revision 2: The Equation (2) used to define the transverse spin state σ is revised as suggested by the Reviewer, only that it's now represented by E_{rr} and $E_{\varphi\varphi}$ instead of E^+ and E^- . We are concerned that introducing E^+ and E^- in the main text will further complicate the presentation of theory. Instead the derivation of Equation (2) from E^+ and E^- is provided in a new supplementary note (**Supplementary Note 3**), and other notes are re-numbered accordingly.

Revision 3: The former definition of transverse spin state we used is now formally defined as the parameter κ , following the Reviewer's notation, to represent the ration of the two cylindrical field components (**in the paragraph before Figure 4**). Using this ratio still shows great convenience in explaining the experimental results, for example the results in Figure 4. **The caption and discussion of Figure 4** are slightly revised by incorporating the parameter κ , including

- In the caption: "... The prediction of $\underline{\kappa}^2$ from numerical calculations is plotted with dashed lines ...".
- In the discussion following Figure 4: "... while the corresponding predicted $\underline{\kappa}^2$ of each device is plotted ..." and "... and the agreement between the ε^2 and $\underline{\kappa}^2$ shows a definitive correspondence ...".

Revision 4: In addition, to obtain the numerical calculations of the new state σ , we calculate κ first and use the dependence of σ on this ratio. This dependence is derived and given in the new **Supplementary Note 3**, and the numerical calculation method for both σ and κ is more clearly explained, and moved **from previously Supplementary Note 1 now to the first section of Methods**. The calculated results of σ for all devices is updated using this method and showed in the new **Figure 3a**.

Revision 5: The **introductory remark of section ‘Transverse spin engineering’** is revised due to the change in σ definition: "The transverse-spin state σ in the waveguide evanescent wave is generally subject to the ratio of longitudinal (E_{long} , along waveguide surface) and transverse (E_{trans} , normal to waveguide surface) field components, $iE_{\text{long}}/E_{\text{trans}}$, in the evanescent region (Supplementary Note 3) ...".

Revision 6: Due to the change of transverse spin definition, other relevant equations are all updated according to the new definition, including **Equations (4), (5), (6), and (7)**. All relevant expressions throughout the paper are modified, including " $\Phi_G = -\sigma\varphi$ " (in the passage after **Equation (5)**), " $S_z = \sigma\hbar$ " (in the first paragraph of section ‘Transverse spin induced SOI’), " $(\pm\sigma_{\text{res}}, \pm l_{\text{res}} \mp \sigma_{\text{res}})$ " (in the first paragraph of section ‘Spin-orbit controlled unidirectional coupling’). The descriptions in Supplementary Notes 4, 5, and 6 are revised accordingly.

Revision 7: The discussion on the geometric phase is moved to **after Equation (4)** to show a better connection with Equation (3). An introductory remark is added before this discussion (the **third sentence after Equation (4)**): "The above interaction between the vector evanescent wave and grating can further be elucidated with the spatial phase acquired by the CVV."

Revision 8: **Figure 1c along with its caption**, used to illustrate the geometric phase, is revised: "... For the CCW WGM shown here, a rotation angle of $\varphi\cdot z$ is experienced by the local coordinates from point (r', φ') to (r'', φ'') , and the geometric phase imparted on the

evanescent wave with a transverse-spin state σ is $\Phi_G = -\sigma\varphi \dots$ ". The previous Supplementary Figure 3 is now removed due to similarity to the new Figure 1c.

Reviewer 2:

The authors have addressed the referees' comments and have considerably improved the presentation in the paper. I still think that notation $2\sigma/(1+\sigma^2)$ should be changed to σ . This is just a notation, and it will considerably simplify the equations. Moreover, the fact that this quantity is the azimuthal phase gradient exactly emphasizes that it is the relevant spin quantity, see PRL 101, 030404 (2008). This is just a minor remark; otherwise, the paper can be published in Nature Communications.

Response: Thanks again for the comment. This change, as well as all relevant revisions needed, have been made as per the response to the comments of Review 1.